# SCALING LAW FOR QUANTIZATION-AWARE TRAINING

## ABSTRACT

Large language models (LLMs) demand substantial computational and memory resources, creating deployment challenges. Quantization-aware training (QAT) addresses these challenges by reducing model precision while maintaining performance. However, the scaling behavior of QAT, especially at 4-bit precision (W4A4), is not well understood. Existing QAT scaling laws often ignore key factors such as the number of training tokens and quantization granularity, which limits their applicability. This paper proposes a unified scaling law for QAT that models quantization error as a function of model size, training data volume, and quantization group size. Through **268** QAT experiments, we show that quantization error decreases as model size increases, but rises with more training tokens and coarser quantization granularity. To identify the sources of W4A4 quantization error, we decompose it into weight and activation components. Both components follow the overall trend of W4A4 quantization error, but with different sensitivities. Specifically, weight quantization error increases more rapidly with more training tokens. Further analysis shows that the activation quantization error in the FC2 layer, caused by outliers, is the primary bottleneck of W4A4 QAT quantization error. By applying mixed-precision quantization to address this bottleneck, we demonstrate that weight and activation quantization errors can converge to similar levels. Additionally, with more training data, weight quantization error eventually exceeds activation quantization error, suggesting that reducing weight quantization error is also important in such scenarios. These findings offer key insights for improving QAT research and development.

## 1 INTRODUCTION

The emergence of large language models (LLMs)(Liu et al., 2024a; Grattafiori et al., 2024; Seed, 2025) revolutionizes natural language processing (NLP), enabling advances in tasks from text generation to complex reasoning. However, their large parameter sizes make them computationally intensive and memory-demanding(Yuan et al., 2024; Zhou et al., 2024), creating challenges for deployment. Quantization (Xiao et al., 2023; Shao et al., 2023), which reduces the precision of model weights and activations, addresses these challenges by lowering memory usage and computational cost. Post-training quantization (PTQ)(Xiao et al., 2023; Ashkboos et al., 2024; Liu et al., 2024b) achieves near-lossless accuracy at moderate precision, such as W8A8 (8-bit weights and activations), but struggles to maintain accuracy at lower precisions like W4A4(Liu et al., 2025a). In contrast, quantization-aware training (QAT) (Chen et al., 2024b; Liu et al., 2025b; Ma et al., 2024; Panferov et al., 2025) incorporates quantization during training, allowing models to adapt to reduced precision and supporting more aggressive compression. However, the scaling behavior of QAT at ultra-low bit-widths (*e.g.*, W4A4) remains underexplored, limiting the design of efficient quantized LLMs.

Scaling laws (Kaplan et al., 2020; Hoffmann et al., 2022) have proven instrumental in understanding LLMs performance as a function of model size, dataset size, and computational resources. Foundational works, such as the Kaplan scaling law (Kaplan et al., 2020) and the refined Chinchilla scaling law (Hoffmann et al., 2022), provide predictive models for optimizing LLM training strategies in full-precision settings. Recent efforts have extended these frameworks to account for model quantization (Kumar et al., 2024; Ouyang et al., 2024; Frantar et al., 2025), with some studies examining PTQ (Kumar et al., 2024; Ouyang et al., 2024) and others proposing QAT-specific scaling laws (Kumar et al., 2024; Frantar et al., 2025). However, existing QAT scaling laws (Kumar et al., 2024;

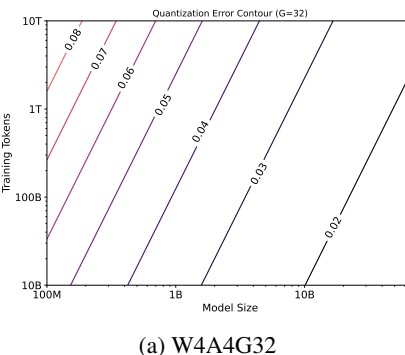 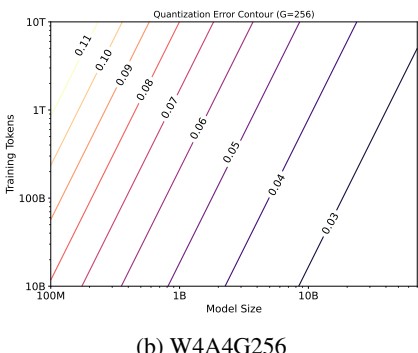

(a) W4A4G32             (b) W4A4G256

Figure 1: **Quantization error contour based on the proposed unified QAT scaling law.** The quantization error decreases as the model size increases, but increases with both the number of training tokens and with coarser quantization granularity.

Frantar et al., 2025) typically focused on either parameters count or fixed quantization settings, often neglecting critical factors such as number of training tokens or the granularity of quantization. Empirical observations (see Figure 4) indicate that quantization error can increase dramatically with larger training datasets and coarser quantization groups. Yet, prior works (Kumar et al., 2024; Frantar et al., 2025) have not provided a unified framework that accounts for the interplay between all these factors, reducing their practical utility for real-world model design and training.

In this paper, we address these limitations by presenting a unified scaling law for QAT. Our model explicitly describes how quantization error depends on model size, the number of training tokens, and quantization granularity. Given that W8A8 quantization achieves nearly lossless performance (Shao et al., 2023; Liu et al., 2024a; Fishman et al., 2024), we focus our analysis on W4A4 QAT. We conduct 268 QAT experiments and show that quantization error decreases as model size increases, but increases with larger training datasets and coarser quantization granularity. Figure 1 shows the contours of quantization error in loss according to our proposed QAT scaling law. Our main contributions are as follows:

- **Unified QAT scaling law**: We propose a mathematical model for QAT quantization error, capturing its dependence on model size, dataset size, and quantization group size.

- **Empirical validation**: Through systematic experiments, we show that quantization error decreases with larger models but increases with more training tokens and coarser quantization.

- **Quantization error decomposition**: We decompose quantization error into weight and activation components, and find that weight quantization error is more sensitive to number of training tokens. We also identify activation quantization—especially in the *FC2* layer of the feed-forward network—as the main bottleneck for W4A4 QAT.

- **Bottleneck layer analysis**: We show that activation quantization error in *FC2* mainly arises from outlier values that 4-bit quantization cannot capture. By keeping the bottleneck layer at 8-bit precision during W4A4 QAT, we demonstrate that weight and activation quantization errors contribute almost equally to the total error at a data-to-parameter ratio of 100. With larger data-to-parameter ratios, weight quantization error surpasses activation error. This highlights the importance of considering both weight and activation components in future QAT algorithm design.

## 2 RELATED WORKS

**Scaling Law of LLMs.** Scaling laws provide a general framework for understanding how model performance changes with resources, guiding both architecture and training strategy design. The Kaplan scaling law (Kaplan et al., 2020) first described how model size, dataset size, and compute relate to performance. Later, the Chinchilla scaling law (Hoffmann et al., 2022) refined this by emphasizing the balance between model parameters (N) and training data tokens (D) for optimal performance under a fixed compute budget. Recent research extends scaling laws to model compression, including quantization (Frantar et al., 2022; Lin et al., 2023). Studies on PTQ scaling

laws (Kumar et al., 2024; Ouyang et al., 2024; Panferov et al., 2025) find that PTQ error decreases as model size increases, but increases with larger training datasets, implying that models trained on more data may need higher precision. Other works on QAT scaling laws (Kumar et al., 2024; Frantar et al., 2025) show that quantization error mainly depends on model size. Building on these studies, our work explores QAT in greater depth and proposes a unified scaling law that considers model size, training data size, and quantization granularity.

**Quantization of LLMs.** Quantization reduces the computational and memory costs of serving LLMs. Current PTQ methods (Xiao et al., 2023) perform well at 8-bit precision, often achieving near-lossless results (*e.g.*, W8A8 quantization). However, lowering the bit-width to 4-bit (*e.g.*, W4A4) with PTQ usually leads to significant performance drops (Shao et al., 2023; Chen et al., 2024a; Liu et al., 2025a). This accuracy loss limits the adoption of efficient 4-bit matrix multiplication (GEMM) kernels (Li et al., 2024) for LLM inference. QAT (Liu et al., 2025b; Chen et al., 2024b) addresses this by training models with quantization applied, which helps recover accuracy at low bit-widths. The BitNet series (Ma et al., 2024; Wang et al., 2024) shows that QAT outperforms PTQ, especially at very low bit-widths, though a gap remains compared to full-precision models. Understanding scaling behavior under QAT is therefore important for designing better QAT strategies.

## 3 PRELIMINARIES

**Classical scaling law.** The Chinchilla scaling law (Hoffmann et al., 2022) models the final loss ($L$) using model size ($N$) and number of training tokens($D$):

$$L(N, D) = \frac{A}{N^\alpha} + \frac{B}{D^\beta} + E, \tag{1}$$

where $A$, $\alpha$, $B$, $\beta$, and $E$ are fitted constants, listed in Table 1. Section E in the Appendix explains the fitting process.

**Existing QAT scaling law.** Previous studies (Frantar et al., 2025; Kumar et al., 2024) modify Eq.(1) by introducing an effective parameter multiplier (EPM) on $N$, resulting in:

$$L(N, D) = \frac{A}{(N \cdot \mathbf{eff}(\mathbf{C}))^\alpha} + \frac{B}{D^\beta} + E, \tag{2}$$

where $\mathbf{eff}(\mathbf{C}) \in [0, 1]$ denotes the EPM, which depends on the model architecture and compression method. A higher value of EPM indicates better preservation of the original (BFloat16 (Kalamkar et al., 2019)) model performance.

**Proposed QAT Scaling Law.** Unlike existing QAT scaling laws that modify the $N$ capacity term in the Chinchilla scaling law, we directly model the final loss gap (*i.e.*, the quantization error) between QAT models and their BFloat16 counterparts . For instance, the quantization error in the EPM scaling law can be calculated through Eq. (2) − Eq. (1):

$$\delta_p(N) = \frac{A}{(N \cdot \mathbf{eff}(\mathbf{C}))^\alpha} - \frac{A}{N^\alpha}, \tag{3}$$

$\delta_p$ represents the quantization error with $p$-bit QAT. Eq. (3) shows that previous QAT scaling laws assume the quantization error depends only on $N$ and is independent of the data size $D$. However, our experiments (Figure 4b) show that the quantization error between W4A4 QAT and BF16 models increases as the data size grows. To address this, we introduce a new quantization error term that depends on both $N$ and $D$. Furthermore, since fine-grained quantization is essential for 4-bit QAT performance (Dettmers & Zettlemoyer, 2023; Rouhani et al., 2023), we also include the quantization granularity $G$ to capture its effect on performance degradation. Thus, our proposed QAT scaling law is:

$$L(N, D, G) = \underbrace{\frac{A}{N^\alpha} + \frac{B}{D^\beta} + E}_{\text{Chinchilla loss}} + \underbrace{\delta_p(N, D, G)}_{\text{low-bit QAT effect}}, \tag{4}$$

where $\delta_p(N, D, G)$ denotes the quantization error for $p$-bit QAT, as a function of $N$, $D$, and $G$. Note that prior scaling laws for PTQ (Kumar et al., 2024) consider both $D$ and $G$ when modeling quantization error. In contrast, existing QAT scaling laws (Kumar et al., 2024; Frantar et al., 2025) consider only $N$ and neglect $D$. We are the first to show that $D$ also affects QAT quantization error.

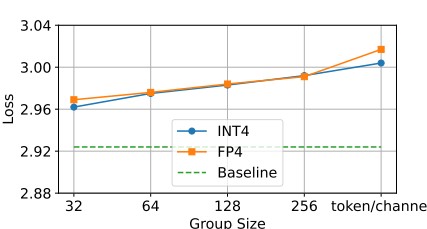 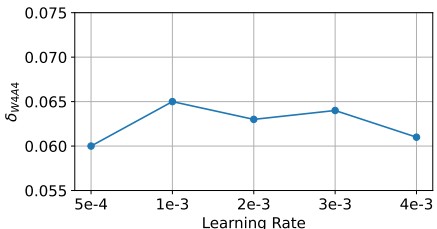

Figure 2: Integer (INT4) vs. floating-point (FP4) in W4A4, 297M model, 50B tokens. INT4 and FP4 achieve similar final loss with QAT.

Figure 3: $\delta_{W4A4}$ at different learning rates, W4A4 ($G = 128$) 145M model, 20B tokens. Quantization error ($\delta_{W4A4}$) maintains in a similar level across different learning rates.

## 4 QAT SCALING LAW

This section introduces a unified scaling law for QAT that incorporates model size $N$, training tokens $D$, and quantization granularity $G$. Section 4.1 outlines the training setups. Section 4.2 presents the main scaling law and reveals an insightful finding distinct from previous studies (Frantar et al., 2025; Kumar et al., 2024) that the number of training tokens $D$ significantly affects QAT error. Section 4.3 analyzes quantization errors from weights and activations separately, identifying activation quantization—especially for the *FC2* layer's input—as the main performance bottleneck. This finding supports a mixed-precision strategy discussed in Section 4.4. Finally, Section 4.5 compares our scaling law with previous approaches.

### 4.1 TRAINING SETUP

**Models and dataset.** We train a series of Llama3-style (Grattafiori et al., 2024) models on the OLMo2-Mix-1124 (OLMo et al., 2024) pretraining dataset. Our experiments systematically explore LLM pretraining across parameter sizes $N \in \{74, 145, 297, 595\}$ million and training token numbers $D \in \{10, 20, 50, 100\}$ billion tokens. For validation purpose, we also train models with 973M parameters on 100 and 200 billion tokens to verify the extrapolation reliability of our scaling law when increasing both model and dataset size. These 268 QAT experiments on A100 GPUs consumed 276K GPU-hours in total. Detailed architectural settings for each model are provided in Sec F.3.

**Evaluation metric.** Following the Chinchilla scaling law (Li et al., 2025), we use the smoothed training loss as an unbiased estimate of validation loss for simplicity and consistency.

**Quantization precision.** Considering that 8-bit can achieve nearly lossless performance (Xiao et al., 2023; Zheng et al., 2025) This work focuses on 4-bit quantization. We train models under three quantization settings: W4A4, W4A16 (only weights quantized to 4-bit), and W16A4 (only activations quantized to 4-bit). The latter two settings help decouple the error sources in W4A4.

**Quantization granularity.** Quantization granularity $G$ refers to the number of elements in each quantization group and is crucial for low-bit quantization (Dettmers & Zettlemoyer, 2023). For each model, we experiment with group sizes $G \in \{32, 64, 128, 256, \text{per-token/channel}\}$.

**Low-precision formats.** Low-bit quantization employs either integer (INT) or floating-point (FP) types. Figure 2 shows that INT4 matches FP4 performance in group-wise quantization and surpasses FP4 by 0.015 in loss for per-channel/token quantization. This advantage stems from INT4's 16 representable values compared to FP4's 15 (Wang et al., 2025), with greater impact in coarse-grained quantization. We adopt the integer format for our scaling law due to its equivalent or superior performance. We hypothesize that INT and FP exhibit similar scaling behavior. Figure 13 verifies that the scaling law fitted to INT4 data also accurately predicts QAT error trend for FP4.

**Training hyper-parameters.** We follow Olmo2 (OLMo et al., 2024) for training hyper-parameters, detailed in Table 4. One key hyper-parameter is the learning rate. For example, BitNet (Ma et al., 2024) shows ternary models benefit from higher learning rates than uncompressed models. In con-

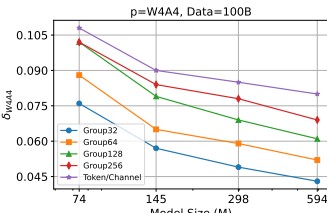 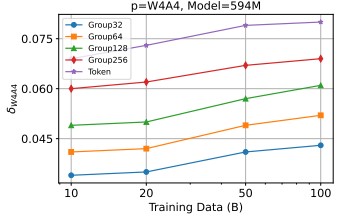 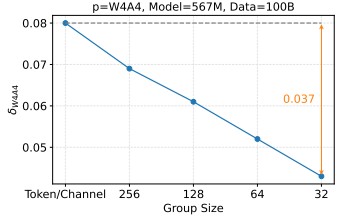

(a) $\delta_{W4A4}$ decreases as model size $N$ increases.

(b) $\delta_{W4A4}$ increases with a greater number of training tokens $D$.

(c) $\delta_{W4A4}$ decreases with smaller group sizes $G$.

Figure 4: **Trend of $\delta_{W4A4}$ with varying $N$, $D$, and $G$.** (a) $\delta_{W4A4}$ decreases as model size increases. (b) $\delta_{W4A4}$ increases with more training tokens. (c) $\delta_{W4A4}$ decreases with smaller group sizes. Note that these trends of $\delta_{W4A4}$ are consistent across different $N$, $D$, and $G$. For simplicity, we merely plot the model trained with 100B tokens in (a), a model size of 594M in (b), and the 594M model trained with 100B tokens in (c).

trast, our focus on 4-bit quantization, which is less aggressive than ternary, leads to less sensitivity to learning rate. We compare uncompressed and W4A4 QAT models, as shown in Figure 3, observe that the quantization error remains nearly constant (within [0.6, 0.65]) across learning rates from $5 \times 10^{-4}$ to $4 \times 10^{-3}$. This indicates that 4-bit QAT does not benefit from higher learning rates compared to uncompressed models. Therefore, we use the same hyper-parameters for both uncompressed and QAT training.

## 4.2 Unified Scaling Law for QAT

**Observation.** The ground truth for $\delta_{W4A4}$ is defined as $loss_{bf16} - loss_{W4A4}$, where $loss_{bf16}$ and $loss_{W4A4}$ denote the final model losses obtained from training with original BFloat16 precision and W4A4 QAT, respectively. To better understand $\delta_{W4A4}$, we plot its relationship with $N$, $D$, and $G$ in Figure 4. We observe three primary trends:

- **Quantization error decrease with increasing model size:** Figure 4a shows that $\delta_{W4A4}$ consistently decreases as model size increases, across different quantization granularities. For example, when model size grows from 74M to 594M, $\delta_{W4A4}$ decreases by an average of 34% across all granularities.

- **Quantization error increase with more training tokens:** Figure 4b indicates that $\delta_{W4A4}$ increases as the number of training tokens grows. Specifically, increasing the training tokens from 10B to 100B results in an average increase of 22% in $\delta_{W4A4}$ across different granularities.

- **Quantization error decrease with finer quantization granularity:** As illustrated in Figure 4c, $\delta_{W4A4}$ decreases as quantization granularity becomes finer. The difference in $\delta_{W4A4}$ between the coarsest and finest quantization granularities is 0.037, which is nearly half the quantization error of the coarsest quantization granularity.

**Proposed scaling law for QAT quantization error.** Existing QAT scaling laws (Kumar et al., 2024; Frantar et al., 2025) account only for model size $N$, overlooking the effects of training data volume $D$ and quantization granularity $G$. To enhance the prediction of QAT quantization error, we propose a comprehensive formula based on our observations:

$$\delta_p(N, D, G) = \frac{k \cdot D^{\gamma_D} \cdot (\log_2(G))^{\gamma_G}}{N^{\gamma_N}}, \tag{5}$$

where $k$, $\gamma_N$, $\gamma_D$ and $\gamma_G > 0$ are fitted parameters. We incorporate a logarithmic term for $G$, as $G = 1$ (no quantization) yields $\delta_p = 0$. The magnitudes of $\gamma_N$, $\gamma_D$ and $\gamma_G$ reflect the sensitivity of the quantization error $\delta_p$ to $N$, $D$ and $G$, respectively. The formula indicates that $\delta_p$ increases with $D$ and $G$ but decreases with $N$.

**Fitting and validation.** We fit Eq.(5) to the ground truth W4A4 quantization error ($\delta_{W4A4}$) obtained from 80 W4A4 QAT runs. Table1 lists the fitted parameters, and Figure 5 compares the actual and predicted $\delta_{W4A4}$. As shown in Figure 5, Eq. (5) accurately models the observed W4A4 QAT


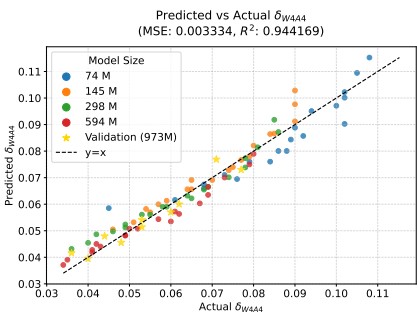
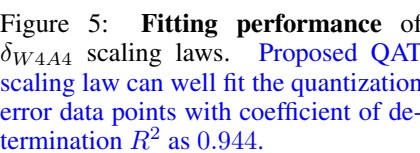
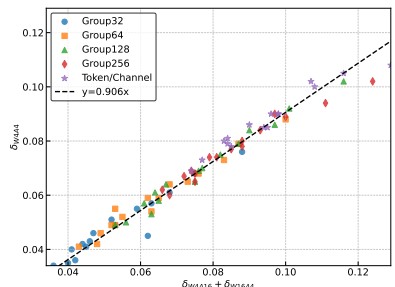

Figure 5: **Fitting performance** of $\delta_{W4A4}$ scaling laws. Proposed QAT scaling law can well fit the quantization error data points with coefficient of determination $R^2$ as 0.944.

Figure 6: **Quantization error decomposition.** $\delta_{W4A4} = k(\delta_{W16A4} + \delta_{W4A16})$ with k=0.906, demonstrating that we can analyze the source of $\delta_{W4A4}$ through exploring $\delta_{W16A4}$ and $\delta_{W4A16}$, respectively.

Table 1: Fitted hyperparameters and their values in our proposed QAT error scaling law.

| Type | Constant | Value | Type | Constant | Value |
|---|---|---|---|---|---|
| Chinchilla | $E$ | 1.9279 | $\delta_{W4A4}$ | $k$ | 0.1582 |
| | $A$ | 237.7042 | | $\gamma_N$ | 0.2186 |
| | $\alpha$ | 0.3022 | | $\gamma_D$ | 0.0745 |
| | $B$ | 596.2490 | | $\gamma_G$ | 0.7779 |
| | $\beta$ | 0.3022 | | | |
| $\delta_{W4A16}$ | $k$ | 0.2522 | $\delta_{W16A4}$ | $k$ | 0.1004 |
| | $\gamma_N$ | 0.3589 | | $\gamma_N$ | 0.1816 |
| | $\gamma_D$ | 0.1610 | | $\gamma_D$ | 0.0331 |
| | $\gamma_G$ | 0.3533 | | $\gamma_G$ | 0.9812 |
| $\delta_{W4A4}$ (*FC2* input 8-bit) | $k$ | 0.3519 | $\delta_{W16A4}$ (*FC2* input 8-bit) | $k$ | 0.1273 |
| | $\gamma_N$ | 0.2637 | | $\gamma_N$ | 0.2347 |
| | $\gamma_D$ | 0.0964 | | $\gamma_D$ | 0.0827 |
| | $\gamma_G$ | 0.3407 | | $\gamma_G$ | 0.4491 |

quantization errors. We further validate the fitted scaling law by predicting the QAT losses of 973M-parameter models trained with $\{100B, 200B\}$ tokens. The consistently accurate predictions indicate that our proposed QAT scaling law generalizes well to larger models and more training data.

### 4.3 DECOMPOSITION OF QUANTIZATION ERROR: WEIGHT VS. ACTIVATION

Although the unified QAT scaling law in Eq. (5) predicts the overall quantization error for W4A4, it remains unclear whether this error mainly arises from weights or activations. Understanding this distinction is essential for targeted optimization. In practice, for a model trained with W4A4 QAT, we cannot directly measure the individual contributions of weight and activation quantization errors. For example, simply disabling quantization in a W4A4 QAT model does not restore the performance of the original unquantized model and may even decrease accuracy further. This occurs because quantization is integrated into the QAT training process, and model parameters adapt to quantization errors during training. To analyze the sources of quantization error in a W4A4 QAT model, we train two additional QAT models: one with W4A16 and another with W16A4.

**Rationale for error decomposition.** As shown in Figure 6, the final quantization error of W4A4 ($\delta_{W4A4}$) can be closely approximated by summing the quantization errors from W4A16 and W16A4 ($\delta_{W4A16} + \delta_{W16A4}$). The observed coefficient between $\delta_{W4A4}$ and $\delta_{W4A16} + \delta_{W16A4}$ is 0.906. This strong correlation suggests that we can effectively analyze $\delta_{W4A4}$ by separately examining the $\delta_{W4A16}$ and $\delta_{W16A4}$.

**How do $\delta_{W4A16}$ and $\delta_{W16A4}$ change with $N$, $D$ and $G$?** Section 4.2 examines how $\delta_{W4A4}$ varies with model size $N$, number of training tokens $D$, and quantization granularity $G$. It is important to

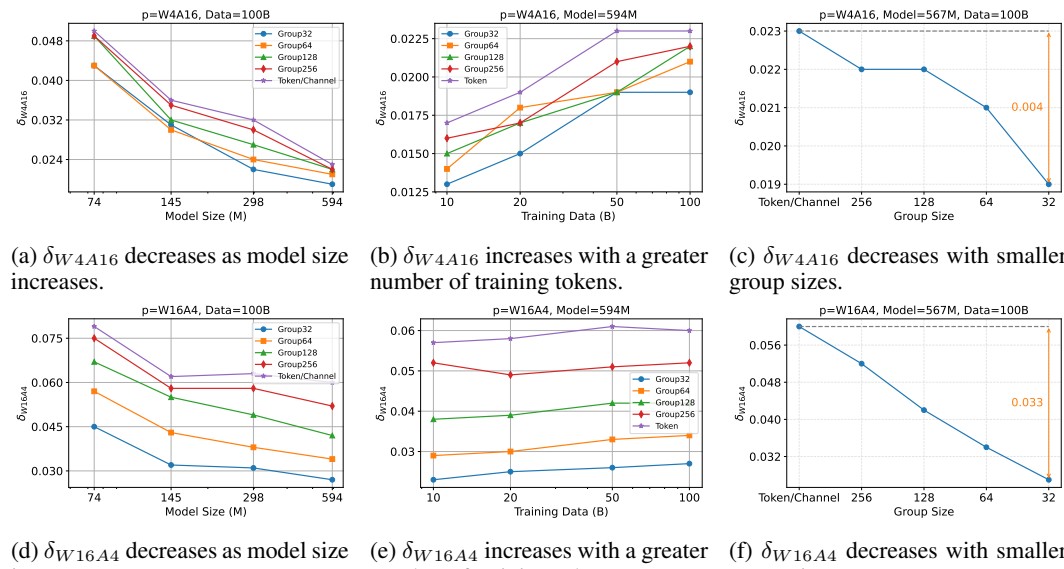

(a) $\delta_{W4A16}$ decreases as model size increases.

(b) $\delta_{W4A16}$ increases with a greater number of training tokens.

(c) $\delta_{W4A16}$ decreases with smaller group sizes.

(d) $\delta_{W16A4}$ decreases as model size increases.

(e) $\delta_{W16A4}$ increases with a greater number of training tokens.

(f) $\delta_{W16A4}$ decreases with smaller group sizes.

Figure 7: (a)-(c) $\delta_{W4A16}$ and (d)-(f) $\delta_{W16A4}$ trend with varying $N$, $D$ and $G$.

see if $\delta_{W4A16}$ and $\delta_{W16A4}$ follow similar patterns. To investigate this, we plot $\delta_{W4A16}$ and $\delta_{W16A4}$ against $N$, $D$ and $G$ in Figure 7, and report the fitted QAT scaling law parameters in Table 1. The results show that both $\delta_{W4A16}$ and $\delta_{W16A4}$ follow trends consistent with $\delta_{W4A4}$, but the degree of sensitivity differs between them:

- $\delta_{W4A16}$ **decreases faster than** $\delta_{W16A4}$ **as model size increases:** The parameter $\gamma_N$ indicates the sensitiveness of quantization error to model size. For $\delta_{W4A16}$, $\gamma_N$ is 0.3589, higher than 0.1816 for $\delta_{W16A4}$. This means weight quantization error decreases more rapidly with larger model size than activation quantization error. As shown in Figure 7 (a) and (d), when model size increases from 74M to 594M, $\delta_{W4A16}$ drops by 51% on average, while $\delta_{W16A4}$ decreases by 34%.

- $\delta_{W4A16}$ **increases faster than** $\delta_{W16A4}$ **as the number of training tokens increases:** The parameter $\gamma_D$ indicates the sensitiveness of quantization error to training tokens. For $\delta_{W4A16}$, $\gamma_D$ is 0.1610, much larger than 0.0331 for $\delta_{W16A4}$. Thus, weight quantization error increases more sharply with more training tokens than activation quantization error. As shown in Figure 7 (b) and (e), increasing training tokens from 10B to 100B raises $\delta_{W4A16}$ by 43% on average, but only increases $\delta_{W16A4}$ by 12%.

- $\delta_{W16A4}$ **is more sensitive to quantization granularity than** $\delta_{W4A16}$**:** The parameter $\gamma_G$ indicates the sensitiveness of quantization error to quantization granularity. For $\delta_{W16A4}$, $\gamma_G$ is 0.9821, much higher than 0.3533 for $\delta_{W4A16}$. This shows that activation quantization error is much more sensitive to granularity, likely due to outliers. As shown in Figure 7 (c) and (f), the gap in $\delta_{W16A4}$ between the coarsest and finest granularity is 0.031, nearly eight times larger than the corresponding gap for $\delta_{W4A16}$.

**Which contributes more to quantization error, $\delta_{W4A16}$ or $\delta_{W16A4}$?** Both weight and activation quantization errors depend on $D$, $N$ and $G$. To compare their contributions, we examine $\delta_{W4A16}$ and $\delta_{W16A4}$ across different parameter values, including fixed data-to-parameter ratios $\frac{D}{N}$, as models with similar $\frac{D}{N}$ often show comparable convergence levels (Hoffmann et al., 2022). Figure 8a shows heatmaps of $R = \frac{\delta_{W16A4}}{\delta_{W4A16}}$. Across all tested $\frac{D}{N}$ ratios and group sizes $G$, $R$ is consistently greater than 1, indicating that activation quantization error generally exceeds weight quantization error. However, the value of $R$ varies with different settings:

- $R$ decreases as $\frac{D}{N}$ increases, because $\delta_{W4A16}$ grows faster with $D$ than $\delta_{W16A4}$. For example, with $G = 32$, $R$ drops from 1.67 at $\frac{D}{N} = 100$ to 1.20 at $\frac{D}{N} = 1000$.

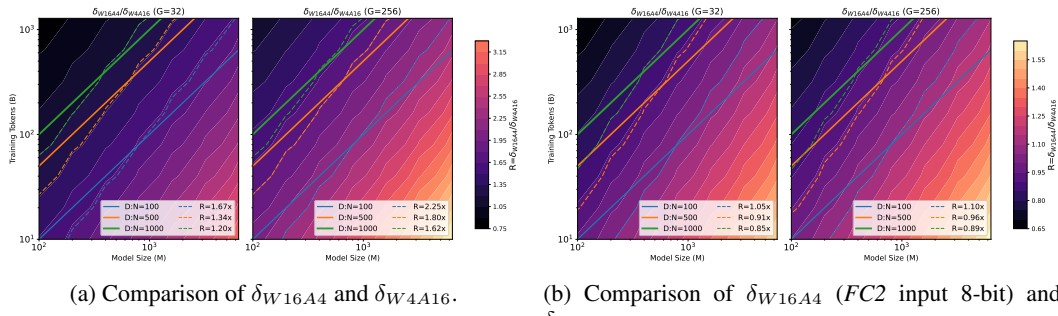

(a) Comparison of $\delta_{W16A4}$ and $\delta_{W4A16}$.

(b) Comparison of $\delta_{W16A4}$ (*FC2* input 8-bit) and $\delta_{W4A16}$.

Figure 8: **Weight and activation quantization errors comparisons.** We report heatmaps of $R = \frac{\delta_{W16A4}}{\delta_{W4A16}}$ across $D$ and $N$, with group sizes 32 and 256. Larger $R$ indicates greater activation quantization error compared to weights.

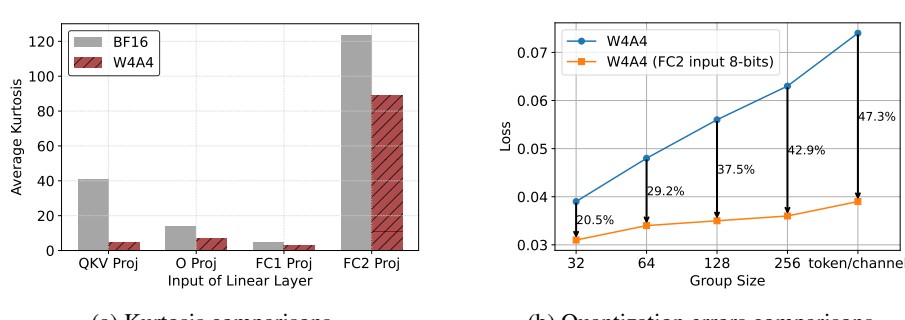

(a) Kurtosis comparisons.

(b) Quantization errors comparisons.

Figure 9: **Comparison of kurtosis and quantization errors.** (a) Kurtosis of input activations across different linear layers. (b) Quantization error comparison with 8-bit *FC2* input. The model size is 595M, the number of training tokens is 100B, and the group size in (a) is 128.

- $R$ increases as group size $G$ increases, since $\delta_{W16A4}$ is more sensitive to quantization granularity. For instance, at $\frac{D}{N} = 1000$, $R$ rises from 1.20 when $G = 32$ to 1.62 when $G = 256$.

**Practical implications.** These results show that as $\frac{D}{N}$ increases, the main source of quantization error shifts from activations to weights. However, $\delta_{W16A4}$ remains larger than $\delta_{W4A16}$ even at high $\frac{D}{N}$ and fine granularity ($G = 32$), and the gap widens with coarser quantization. Therefore, activation quantization error is usually the dominant factor in W4A4 quantization (as $R > 1$), highlighting the importance of optimizing activation quantization to improve W4A4 QAT performance.

### 4.4 MITIGATING ACTIVATION QUANTIZATION ERROR IN *FC2* PROJ INPUT

Since activation quantization error is the main bottleneck in W4A4 QAT, as shown in the previous section, it is important to understand why activations are harder to quantize than weights and how to address this issue. A major reason is the presence of outliers in large language models, which make activation quantization more difficult (Xiao et al., 2023). This problem is well known in post-training quantization (PTQ), where outliers can cause significant performance drops. Although QAT applies quantization during the entire training process and acts as a regularizer to suppress activation outliers (Nrusimha et al., 2024), some challenges remain, especially in certain layers.

**Persistent outliers in *FC2* Proj input with QAT.** Kurtosis (DeCarlo, 1997; Liu et al., 2024b; Nrusimha et al., 2024) measures the "tailedness" of a distribution, with higher values indicating more outliers. Figure 9a shows that QAT effectively reduces outliers in the input activations of the *QKV* Proj, *O* Proj, and *FC1* Proj layers, so further outlier suppression is not needed for these layers. However, even though QAT lowers the kurtosis of the *FC2* Proj input from 123 to 89, this value

Table 2: **Comparison with other scaling laws.** "Num" indicates the number of scaling laws fitted. "Relative Error" represents the difference between the predicted and actual quantization errors.

| Method | $N$ | $D$ | $G$ | $\delta_p$ | Precision | Num. of $\delta_p$ | Relative Error |
|---|---|---|---|---|---|---|---|
| (Frantar et al., 2025)(Kumar et al., 2024) | ✓ | ✗ | ✗ | Eq. (3) | W4A16 W4A4 | 5 5 | 19.3% 8.5% |
| Ours | ✓ | ✓ | ✓ | Eq. (5) | W4A16 W4A4 | 1 1 | **5.2%** **4.7%** |

is still significantly higher than in other layers. The high kurtosis means that the *FC2* Proj input remains prone to large quantization errors, making it a key contributor to the activation quantization bottleneck described in Sec. 4.3. The main reason for this high kurtosis is that the *FC2* Proj input comes from the output of the SwiGLU (Shazeer, 2020) module. The gating mechanism and non-linear transformations in SwiGLU create a complex activation distribution that amplifies outliers (Zhang et al., 2025). As a result, even with QAT regularization, the *FC2* Proj input remains sensitive to outliers and is the main source of activation quantization error in W4A4 QAT models.

**Mixed-precision approach.** To study the W4A4 scaling law without the activation bottleneck, it is necessary to reduce quantization error in the *FC2* Proj input. This can be achieved by using higher quantization precision or outlier suppression strategies (Xiao et al., 2023; Ashkboos et al., 2024). Since 8-bit quantization achieves near-lossless training (Liu et al., 2024a), we use a simple approach: quantizing the *FC2* Proj input to 8 -bit (denoted as "*FC2* input 8-bit"). While other outlier suppression methods (Xiao et al., 2023; Ashkboos et al., 2024; Chen et al., 2024a) could also be considered, 8-bit quantization provides an upper bound on the improvements possible. This approach offers a general and robust baseline for understanding the potential of the W4A4 QAT scaling law without the activation bottleneck.

**Impact on quantization error.** Figure 9b shows that using 8-bit *FC2* inputs significantly reduces quantization error, especially for coarse-grained quantization, which is more sensitive to outliers. For example, with W4A4 QAT, 8-bit *FC2* lowers quantization error by 20.5% for $G = 32$ and by 42.9% for $G = 256$. This demonstrates that 8-bit *FC2* Proj inputs effectively reduce both the overall activation quantization error and its sensitivity to granularity. Table 1 further supports this, showing that the parameter $\gamma_G$ for $\delta_{W16A4}$ decreases from 0.9812 to 0.4471 when using 8-bit *FC2* Proj inputs. Figure 8b illustrates that, under 8-bit *FC2* inputs, $\delta_{W16A4}$ and $\delta_{W4A16}$ become similar in magnitude, with their ratio $R$ ranging from 0.85 to 1.10 for $\frac{D}{N}$ ratios between 100 and 1000, and for group sizes $G = 32$ and $G = 256$.

**Practical implications.** For practitioners, the main takeaway is that special treatment of the *FC2* Proj input—through mixed-precision quantization or targeted outlier suppression—is crucial for maximizing low-bit QAT performance. Once the *FC2* Proj input bottleneck is removed, further improvements to W4A4 QAT should focus on jointly optimizing both weight and activation quantization errors, as their effects become similar. This suggests a shift in QAT development from mainly activation-focused methods (Panferov et al., 2025; Xiao et al., 2023) to approaches that balance both error sources.

### 4.5 COMPARISONS WITH OTHER QAT SCALING LAWS

We compare our proposed QAT scaling law (Eq. (5)) with existing scaling laws (Frantar et al., 2025; Kumar et al., 2024). Previous methods (Kumar et al., 2024; Frantar et al., 2025) do not account for quantization granularity $G$, so they require separate curves for each $G \in \{32, 64, 128, 256, \text{per-channel/token}\}$ for fair comparison. In contrast, our scaling law models different granularities with a single curve. As shown in Table 2, our approach reduces the relative error from 19.3% to 5.2% for W4A16 QAT and from 8.5% to 4.7% for W4A4 QAT. The larger improvement for W4A16 is due to $\delta_{W4A16}$ increasing more rapidly with $D$ than $\delta_{W16A4}$. Overall, including $D$ in $\delta_p$ improves prediction accuracy, and modeling $G$ increases adaptability to different quantization granularities.

## 5 CONCLUSIONS

This paper proposes a comprehensive scaling law for 4-bit QAT of LLMs, integrating model size, training dataset size, and quantization granularity. The new QAT scaling law is more practical, as it jointly models $N$, $G$, and $D$, and achieves more accurate predictions than previous approaches. We also show that processing the *FC2* input with 8-bit in W4A4 QAT significantly reduces both quantization error and sensitivity to quantization granularity. Furthermore, our analysis shows that, after applying 8-bit quantization to the *FC2* input in W4A4 QAT, weight and activation quantization errors contribute almost equally to the total error. This result suggests that future QAT algorithms should also investigate weight quantization error, rather than focusing solely on activation outliers as previous methods do.

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

## A  OUTLINES

- Sec. D: We justify the focus on 4-bit quantization and show that the proposed scaling law generalizes to other bit widths, including 2-bit and 3-bit.
- Sec. E: We offer more details on fitting the Chinchilla scaling law and the proposed QAT scaling law.
- Sec. F: We provide technical details on quantization formats (INT, FP), quantizer methods (e.g., AbsMax, LAC), and the model architecture.
- Sec. G: We derive the quantization-error contours shown in Figure 1.
- Sec. H: We extend the analysis to the Efficient Parameter Multiplier (EPM).
- Sec. I: We present additional ablation studies on the scaling-law parameters and Hadamard rotation.

## B  LIMITATIONS

This paper proposes a unified QAT scaling law and primarily focuses on experiments with 4-bit dense models. One limitation is that we do not conduct experiments on the MoE (Cai et al., 2025) architecture. Since MoE models contain more weight parameters but similar activation sizes, they may exhibit a different ratio of weight to activation quantization error compared to dense models. Additionally, our analysis mainly centers on W4A4 quantization. While some recent works explore extremely low-bit QAT, such as ternary quantization (Ma et al., 2024; Panferov et al., 2025), investigating unified scaling laws for these settings is also valuable. Finally, the largest training compute consumed for our proposed QAT scaling law in this study is to train a 595M parameter model trained over 100B tokens. Intuitively, the accuracy of scaling law extrapolation would be further improved by increasing both the model size and the number of training tokens.

## C  THE USE OF LARGE LANGUAGE MODELS

We use LLMs to polish the paper, correct the grammar, and for some of the figures in the article, the initial drawing codes are generated by LLMs.

## D  EXTEND TO OTHER BIT-WIDTH

**Main scope.** We deliberately focus on 4-bit quantization for the following reasons: (1) 8-bit quantization is typically near-lossless, making it less critical for analyzing scaling-law errors; (2) 5–7-bit formats currently have limited native hardware support, constraining their practical deployment; and (3) 4-bit quantization is both practical and widely used, and it introduces non-negligible quantization error. Because 4-bit already induces substantial error, we first conduct an in-depth analysis of its bottlenecks—specifically disentangling weight vs. activation errors—before moving to more extreme bit-widths (e.g., 2-bit). This focused approach establishes a clear baseline for understanding quantization effects in realistic settings.

**Other bit-widths.** Some edge devices with tight memory budgets motivate even more aggressive weight quantization (2 or 3-bits). To test how our scaling law generalizes, we additionally evaluate 2-bit and 3-bit *weight-only* quantization. The core principle of our law is the relationship between quantization-induced error and the model size ($D$), number of training tokens ($N$), and group size ($G$; number of channels per scaling group). Consistent with our 4-bit findings, Table 3 shows that these more extreme bit-widths follow the same predictable trends: error increases as training tokens $N$ increase; error decreases as model size $D$ increases; and error decreases as group size $G$ decreases (i.e., more, smaller groups). These results indicate that our scaling law and analysis generalize well across bit-widths.

## E  SCALING LAW FITTING

**Chinchilla Scaling Law.** Our QAT scaling law builds on the classical Chinchilla scaling law (Hoffmann et al., 2022), as defined in Eq. (1). Following the original methodology (Hoffmann et al.,

Table 3: Quantization error in 2-bit and 3-bit quantization across different model size, training tokens and quantization group size.

| Model size | Training/Tokens | Precision | Loss | Quantization error |
|---|---|---|---|---|
| 74M | 10B | Bf16 | 3.294 | - |
| 74M | 20B | Bf16 | 3.231 | - |
| 74M | 50B | Bf16 | 3.169 | - |
| 74M | 100B | Bf16 | 3.153 | - |
| 145M | 10B | Bf16 | 3.207 | - |
| 284M | 10B | Bf16 | 3.099 | - |
| 74M | 10B | W3A16G128 | 3.37 | 0.076 |
| 74M | 20B | W3A16G128 | 3.314 | 0.083 |
| 74M | 50B | W3A16G128 | 3.275 | 0.106 |
| 74M | 10B | W3A16G128 | 3.38 | 0.086 |
| 145M | 10B | W3A16G128 | 3.287 | 0.08 |
| 284M | 10B | W3A16G128 | 3.165 | 0.066 |
| 74M | 10B | W3A16G256 | 3.38 | 0.086 |
| 74M | 10B | W3A16G128 | 3.379 | 0.085 |
| 74M | 10B | W3A16G64 | 3.377 | 0.083 |
| 74M | 10B | W2A16G128 | 3.553 | 0.259 |
| 74M | 20B | W2A16G128 | 3.509 | 0.278 |
| 74M | 50B | W2A16G128 | 3.466 | 0.297 |
| 74M | 10B | W2A16G128 | 3.553 | 0.259 |
| 145M | 10B | W2A16G128 | 3.406 | 0.199 |
| 284M | 10B | W2A16G128 | 3.255 | 0.156 |
| 74M | 10B | W2A16G256 | 3.553 | 0.259 |
| 74M | 10B | W2A16G128 | 3.549 | 0.255 |
| 74M | 10B | W2A16G64 | 3.547 | 0.253 |

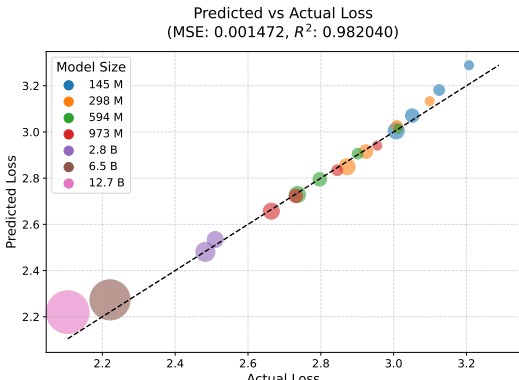

Figure 10: **Fitting performance of chinchilla scaling laws.** The size of the data point is proportional to training data size $D$.

2022), we estimate the parameters ($E$, $A$, $\alpha$, $B$, $\beta$) by minimizing the Huber loss (Huber, 1964) between the predicted and observed log losses, using the L-BFGS algorithm (Goldfarb, 1970). Chinchilla scaling law (Hoffmann et al., 2022) observes that the scaling exponents $\alpha$ and $\beta$ are approximately equal, which suggests that one should scale $N$ and $D$ equally as compute increases. Therefore, we also set $\alpha = \beta$, in line with previous studies (Gadre et al., 2024; Kumar et al., 2024). For our experiments, we train models with sizes ranging from 145M to 2.8B parameters. To improve the extrapolation of the scaling law fit, we include 6.5B and 12.7B parameter models, which we obtain from the official OLMO-2-7B[1] and OLMO-2-13B[2] releases. As shown in Figure 10, the empirical training losses closely match the predicted losses, achieving a mean squared error (MSE)

---

[1]https://huggingface.co/allenai/OLMo-2-1124-7B

[2]https://huggingface.co/allenai/OLMo-2-1124-13B

of 0.0014 and an $R^2$ of 0.982, which indicates a highly accurate fit. It is important to note that our proposed QAT scaling law (Eq. (5)) directly models the quantization error. As a result, it is compatible with any scaling law related to the final loss (Hoffmann et al., 2022; Gadre et al., 2024; Kaplan et al., 2020). In this paper, we choose to use the Chinchilla scaling law for consistency with previous QAT scaling law studies (Kumar et al., 2024; Frantar et al., 2025).

**Proposed Scaling Law Across Different Precisions.** Figure 5 in the main paper illustrates the fitting performance of the proposed scaling law (Eq.(5)) in the W4A4 precision setting. In this section, we further present the fitting results for W4A16 and W16A4 precisions in Figure11, which achieve mean squared errors (MSE) of 0.001 and 0.003, respectively. These results demonstrate the effectiveness of the proposed unified QAT scaling law across different precision configurations. Additionally, we show the fitting performance for W16A4 and W4A4 precisions with the *FC2* input quantized to 8-bit in Figure 12.

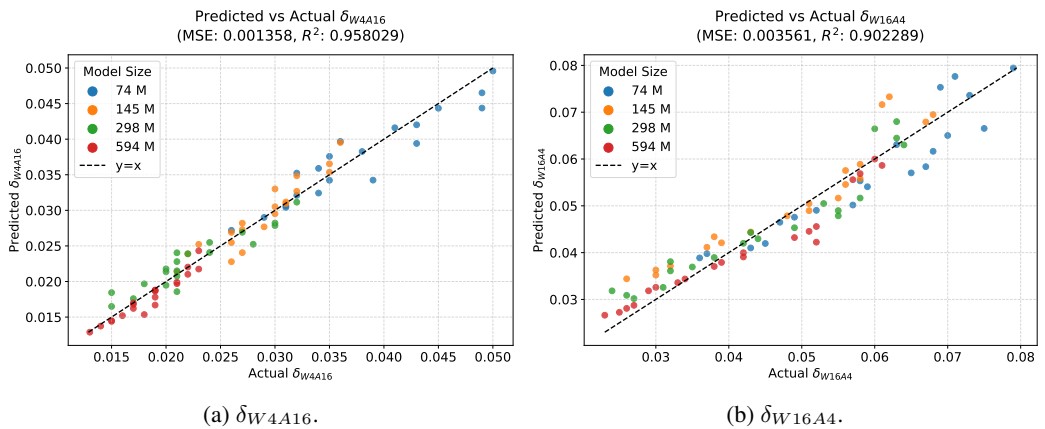

(a) $\delta_{W4A16}$.          (b) $\delta_{W16A4}$.

Figure 11: Fitting performance of proposed scaling law on $\delta_{W4A16}$ and $\delta_{W16A4}$.

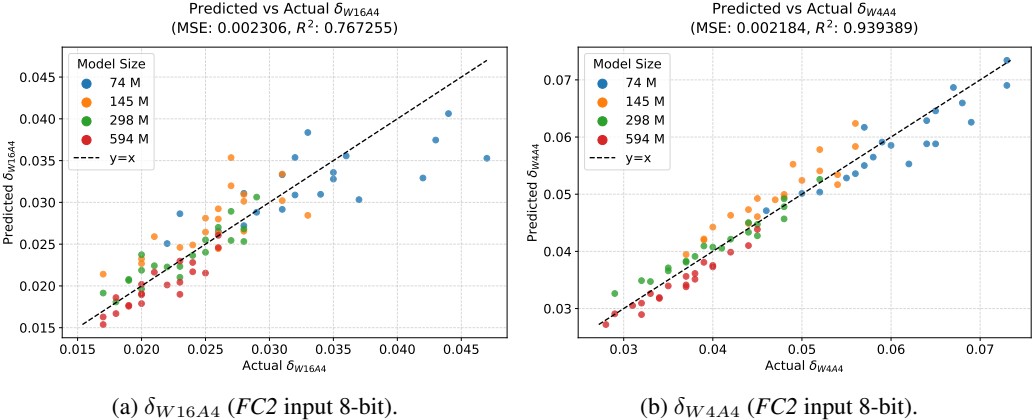

(a) $\delta_{W16A4}$ (*FC2* input 8-bit).          (b) $\delta_{W4A4}$ (*FC2* input 8-bit).

Figure 12: Fitting performance of proposed scaling laws on $\delta_{W16A4}$ and $\delta_{W4A4}$ scaling laws with *FC2* Proj inputs as 8-bit.

# F QUANTIZATION IMPLEMENTATION DETAILS AND TYPES

## F.1 QUANTIZATION TYPES

There are two main types of model quantization: integer (INT) and floating-point (FP) quantization.

**Integer Quantization.** In integer quantization, continuous values are uniformly mapped to discrete integer values. Mathematically, for a given matrix $\mathbf{X}$, the quantization process is defined as:

$$\mathbf{X}_{\text{INT}} = \text{clamp}\left(\lfloor\frac{\mathbf{X}}{s}\rceil, Q_{min}, Q_{max}\right) \tag{6}$$

where $\lfloor\cdot\rceil$ denotes the rounding operation, and $s$ is the scaling factor. Here, $\mathbf{X}_{\text{INT}}$ represents the quantized integer tensor, and $\mathbf{X}$ denotes the original full-precision tensor. After rounding, a clipping operation ensures that the quantized values remain within the range $[Qmin, Q_{max}]$, where $Q_{min} = -2^{b-1}$ and $Q_{max} = 2^{b-1} - 1$, with $b$ being the number of quantization bits. To recover an approximate real value, the quantized tensor can be dequantized by multiplying by the scaling factor $s$:

$$\hat{\mathbf{X}} = \mathbf{X}_{\text{INT}} \times s, \tag{7}$$

**Floating-Point Quantization.** Floating-point representation is more complex than the integer format. Each floating-point number consists of three components: the sign bit ($S$), the exponent ($E$), and the mantissa ($M$). This format is typically denoted as ExMy, where $x$ and $y$ indicate the number of bits allocated to the exponent and mantissa, respectively. The sign bit determines whether the number is positive or negative. The exponent defines the range of representable values, while the mantissa determines the precision. A floating-point number is decoded as:

$$\texttt{Value} = (-1)^S \times (1.M) \times 2^{E-\text{bias}} \tag{8}$$

In this paper, we focus on 4-bit quantization and adopt the E2M1 FP4 format, following previous works (Wang et al., 2025; Sun et al., 2025). For a given matrix $\mathbf{X}$, the quantization process is:

$$\mathbf{X}_{\text{FP}} = \text{MAP}\left(\frac{\mathbf{X}}{s}\right), \tag{9}$$

where $s$ is the scaling factor for normalization, and $\text{MAP}()$ denotes mapping the normalized values to the nearest floating-point values defined by Eq. (8). Similar to integer quantization, the values can be dequantized to approximate real values by multiplying by $s$:

$$\hat{\mathbf{X}} = \mathbf{X}_{\text{FP}} \times s, \tag{10}$$

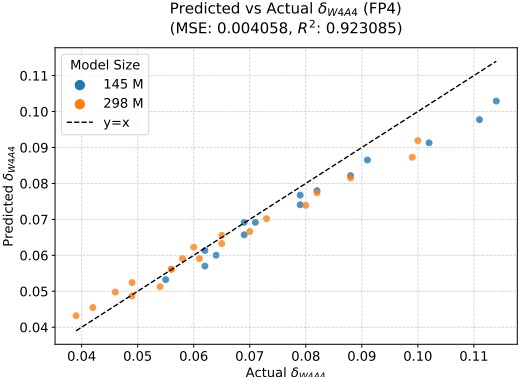

Figure 13: The QAT scaling law, fitted for INT4 quantization, also accurately models the quantization error of FP4 quantization.

**Scaling Behavior.** Consistent with previous work (Kumar et al., 2024), we hypothesize that the scaling behavior for INT and FP formats can be described by the same functional form. There are two pieces of evidence supporting this assumption. First, Figure 2 shows that the performance gap between FP4 and INT4 is negligible in the 4-bit setting. Second, Figure 13 demonstrates that the scaling law fitted on INT4 data also accurately predicts QAT error for FP4.

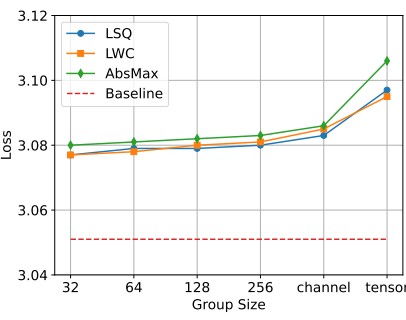 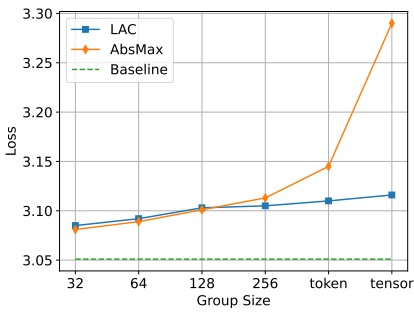

(a) Weight quantizer ablation.   (b) Activation quantizer ablation.

Figure 14: Quantizer ablation studies for 145M model with 50B tokens.

### F.2 QUANTIZER

The quantization format defines the representation space for discrete values. Both integer (INT) and floating-point (FP) formats require a scaling factor to normalize continuous values into a discrete range. Different quantizers employ distinct methods to compute the scaling factor $s$, which is shared within a quantization group. For simplicity, we consider $\mathcal{X}$ as a quantization group here.

**AbsMax.** The AbsMax quantizer computes the scaling factor using the absolute maximum value, given by $\frac{M}{\max(|X|)}$, where $M$ represents the maximum discrete value (e.g., $M = 8$ for INT4, $M = 6$ for E2M1 FP4).

**LWC and LAC.** The LWC (Shao et al., 2023) and LAC (Chen et al., 2024a) quantizers extend AbsMax by introducing learnable clipping factors for weight and activation quantization, respectively. Their scaling factor is computed as $\frac{M}{\max(|X|)\cdot\gamma}$, where $\gamma$ is a learnable clipping factor. LWC assigns a unique $\gamma$ per weight group, while LAC shares $\gamma$ across the same group index for different tokens to enhance deployability.

**LSQ.** The LSQ (Esser et al., 2019) quantizer treats the scaling factor as a directly learnable parameter.

**Ablation of different quantizer.** As shown in Figure 14, activation quantization is more sensitive to quantizer choice than weight quantization, primarily due to outliers in activation distributions (An et al., 2025). For example, all three weight quantizers achieve similar final loss, with differences less than 0.003 across most granularities except per-tensor. Thus, we set the weight quantizer to AbsMax, as we do not use per-tensor quantization. However, for activations, LAC significantly outperforms AbsMax when group size exceeds 256. Therefore, we use AbsMax for activation quantization with fine group sizes ($< 256$), and LAC for activations with coarse group sizes ($\geq 256$).

### F.3 MODEL ARCHITECTURE

We select the Llama-3 (Grattafiori et al., 2024) style model for our experiments due to its wide adoption. As shown in Figure 15, each transformer block in the Llama-3 style model contains four linear layers: *QKV* Proj, *O* Proj, *FC1* Proj, and *FC2* Proj. Additionally, the Llama-3 style model employs Group Query Attention (GQA)(Ainslie et al., 2023) for the self-attention module and SwiGLU(Shazeer, 2020) for the feed-forward module. Table 4 presents the detailed architectural settings of the models used.

## G QUANTIZATION ERROR CONTOUR

Figure 1 shows the contour plot of W4A4 QAT quantization using the proposed QAT scaling law in Eq. (5). For clarity, we restate Eq. (5):

$$\delta_p(N, D, G) = \frac{k \cdot D^{\gamma_D} \cdot (\log_2(G))^{\gamma_G}}{N^{\gamma_N}}.$$

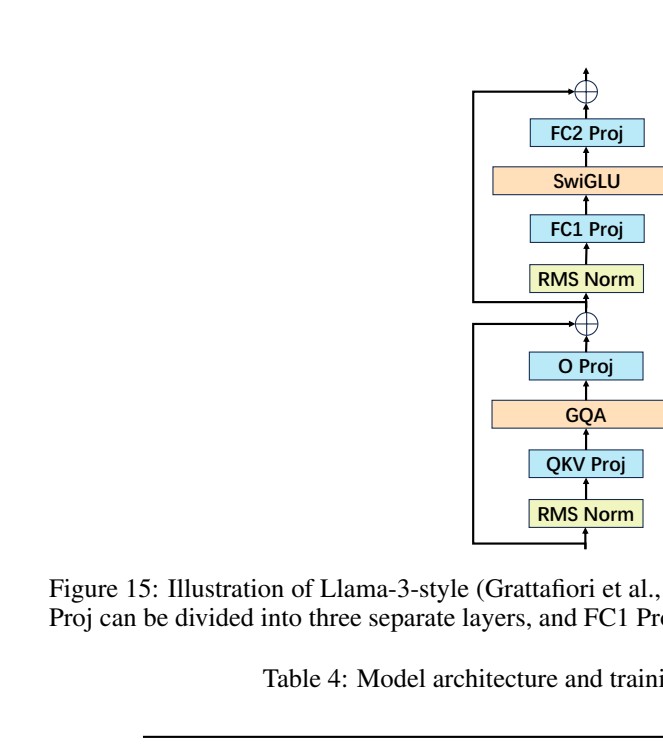

Figure 15: Illustration of Llama-3-style (Grattafiori et al., 2024) transformer block. Note that QKV Proj can be divided into three separate layers, and FC1 Proj can be split into two layers.

Table 4: Model architecture and training hyper-parameters.

| Model Size | 74M | 145M | 297M | 595M | 973M | 2.8B |
|---|---|---|---|---|---|---|
| Layers | 12 | 12 | 12 | 24 | 16 | 28 |
| Hidden Size | 768 | 1024 | 1536 | 1536 | 2048 | 3072 |
| FFN Hidden Size | 2048 | 3072 | 4096 | 4096 | 8192 | 8192 |
| Attention Heads | 16 | 16 | 24 | 24 | 32 | 24 |
| KV Heads | 4 | 4 | 6 | 6 | 8 | 8 |
| Batch Size (# Sequence) | 256 | 256 | 512 | 512 | 512 | 512 |
| Max LR | 1.5e-3 | 1.0e-3 | 8e-4 | 6e-4 | 6e-4 | 6e-4 |
| Min LR | \multicolumn{6}{c}{$0.1 \times$ Max LR} | | | | | |
| Optimizer | \multicolumn{6}{c}{AdamW ($\beta_1 = 0.9, \beta_2 = 0.95$)} | | | | | |
| Weight Decay | \multicolumn{6}{c}{0.1} | | | | | |
| Clip Grad Norm | \multicolumn{6}{c}{1.0} | | | | | |
| LR Schedule | \multicolumn{6}{c}{Cosine} | | | | | |
| Warmup Steps | \multicolumn{6}{c}{500} | | | | | |
| Sequence Length | \multicolumn{6}{c}{2048} | | | | | |

We plot the contour by fixing $G$. Let $C = k \cdot (\log_2(G))^{\gamma_G}$, so Eq. (5) simplifies to:

$$\delta_p(N, D, G) = C \cdot D^{\gamma_D} \cdot N^{-\gamma_N}.$$

Each contour line represents a constant quantization error, i.e., $\delta_p(N, D) = z_0$:

$$C \cdot D^{\gamma_D} \cdot N^{-\gamma_N} = z_0.$$

Taking the base-10 logarithm of both sides, we have:

$$\log_{10}(C) + \gamma_D \log_{10}(D) - \gamma_N \log_{10}(N) = \log_{10}(z_0)$$
$$\gamma_D \log_{10}(D) - \gamma_N \log_{10}(N) = \log_{10}(z_0) - \log_{10}(C)$$
$$\gamma_D \log_{10}(D) + (-\gamma_N) \log_{10}(N) = \text{const}$$

Let $x = \log_{10}(N)$ and $y = \log_{10}(D)$. The contour equation becomes:

$$\gamma_D y - \gamma_N x = \text{const}$$

or equivalently,

$$y = \frac{\gamma_N}{\gamma_D} x + \text{const}'$$

Thus, in the $(\log_{10} N, \log_{10} D)$ space, the contours are straight lines. The slope of each contour line is $\frac{\gamma_N}{\gamma_D}$.

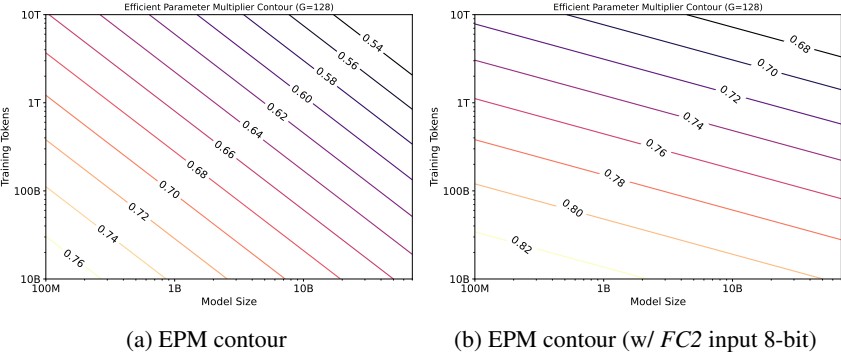

(a) EPM contour

(b) EPM contour (w/ *FC2* input 8-bit)

Figure 16: **Efficient parameter multiplier (EPM) contour for W4A4 QAT.** EPM of W4W4 QAT consistently outperform $0.5$, and setting *FC2* inputs as 8bit significantly improve the EPM with

## H    SCALING WITH EFFICIENT PARAMETER MULTIPLIER

To improve the practicality of the proposed QAT scaling law, we extend it to the efficient parameter multiplier (EPM) (Eq. (2)) (Frantar et al., 2025; Kumar et al., 2024), which quantifies the impact of quantization on the model's effective parameter count. Previous studies (Frantar et al., 2025; Kumar et al., 2024) treat eff($C$) as a constant determined by the model architecture and quantization type, independent of model size and the number of training tokens. In contrast, we model the quantization error $\delta_p$ instead of directly modeling eff($C$). However, we can derive the value of eff($C$) by solving the following equation:

$$\underbrace{\frac{A}{N^\alpha} + \frac{B}{D^\beta} + E + \delta_p(N, D, G)}_{\text{Loss with QAT (Eq. (4))}} = \underbrace{\frac{A}{(N \cdot \mathbf{eff}(\mathbf{C}))^\alpha} + \frac{B}{D^\beta} + E}_{\text{Loss without QAT (Eq. (2))}}. \tag{11}$$

From this, we obtain:

$$\mathbf{eff}(\mathbf{C}) = \left( \frac{A}{A + \delta_p(N, D, G) \cdot N^\alpha} \right)^{\frac{1}{\alpha}}. \tag{12}$$

By substituting $\delta_p$ with Eq. (5), the final expression for eff($C$) is:

$$\mathbf{eff}(\mathbf{C}) = \left( \frac{A}{A + k \cdot D^{\gamma_D} \cdot (\log_2(G))^{\gamma_G} \cdot N^{\alpha - \gamma_N}} \right)^{\frac{1}{\alpha}}, \tag{13}$$

where $N$, $D$, and $G$ are variables, and $A$, $k$, $\alpha$, $\gamma_D$, $\gamma_G$, and $\gamma_N$ are constants. Eq. (13) shows that eff($\mathbf{C}$) decreases as $D$ and $G$ increase. Furthermore, the relationship between eff($\mathbf{C}$) and $N$ depends on the difference $\alpha - \gamma_N$. Although the quantization error decreases as the model size increases, with $\gamma_N$ indicating the rate of this decrease, the speed at which the loss decreases also slows down with larger model sizes, as represented by $\alpha$. This explains why the relationship between EPM and $N$ depends on $\alpha - \gamma_N$. Since $\alpha > \gamma_N$ in the W4A4 scenario (as shown in Table 1), eff($\mathbf{C}$) also decreases as $N$ increases. As shown in Figure 16a, the EPM for W4A4 exceeds $0.5$ in most cases, indicating that W4A4 QAT achieves a better trade-off than even lossless W8A8. Additionally, Figure 16b demonstrates that setting the *FC2* input to 8 bits significantly improves EPM, increasing it by $0.06$ to $0.14$ across different values of $N$ and $D$.

**Practical implications.** Our results show that EPM is sensitive to model size, training data, and quantization granularity. EPM serves as a practical metric for evaluating the effective capacity of quantized models under different settings. It also helps predict when resource-intensive quantization methods, such as fine-grained or mixed-precision quantization, are worthwhile. While these methods can improve EPM, they also increase inference overhead. EPM therefore helps balance the trade-off between higher effective capacity and additional computational cost.

## I   MORE ABLATION STUDIES

Table 5: Ablation study of incorporating $D$ in Eq. (5) across various precisions.

| Precision | Ablation | Relative Error |
|-----------|----------|----------------|
| W4A4 | w/o $D$ | 8.6% |
|  | w/ $D$ | 4.7% |
| W4A16 | w/o $D$ | 13.8% |
|  | w/ $D$ | 5.2% |

Table 6: Ablation studies about random hadamard rotation. Models in this table are 145M parameters with 20B training tokens.

| Group size | Precision | Outlier | Final training loss | Quantization error |
|------------|-----------|---------|---------------------|--------------------|
| - | Bf16 | - | 3.125 | - |
| channel/token | W4A4 | - | 3.209 | 0.084 |
| channel/token | W4A4 | FC2 input rotation | 3.178 | 0.053 |
| channel/token | W4A4 | FC2 input 8-bit | 3.173 | **0.048** |
| 256 | W4A4 | - | 3.196 | 0.071 |
| 256 | W4A4 | FC2 input rotation | 3.174 | 0.049 |
| 256 | W4A4 | FC2 input 8-bit | 3.167 | **0.042** |
| 128 | W4A4 | - | 3.19 | 0.065 |
| 128 | W4A4 | FC2 input rotation | 3.171 | 0.046 |
| 128 | W4A4 | FC2 input 8-bit | 3.165 | **0.04** |
| 64 | W4A4 | - | 3.18 | 0.055 |
| 64 | W4A4 | FC2 input rotation | 3.169 | 0.044 |
| 64 | W4A4 | FC2 input 8-bit | 3.164 | **0.039** |
| 32 | W4A4 | - | 3.172 | 0.047 |
| 32 | W4A4 | FC2 input rotation | 3.165 | 0.04 |
| 32 | W4A4 | FC2 input 8-bit | 3.16 | **0.035** |

**Ablation studies about $D$.** The main difference between our scaling law and existing methods (Kumar et al., 2024; Frantar et al., 2025) is that we recognize $\delta_p$ increases with $D$ and explicitly include $D$ in the scaling law. Table 5 shows ablation results for removing $D$ from Eq. (5). Excluding $D$ reduces prediction accuracy for both W4A4 and W4A16: the relative error for W4A4 rises from 4.7% to 8.6%, and for W4A16 from 5.2% to 13.8%. These results highlight the necessity of including $D$ in the QAT scaling law.

**Ablation studies about Hadamard rotation.** The activation-quantization error bottleneck lies in the FC2 inputs. We therefore examine an outlier-mitigation technique for this bottleneck. As shown in Table 6, we apply a random Hadamard rotation Ashkboos et al. (2024) to the FC2 inputs and the corresponding inverse rotation to the FC2 weights to preserve computational equivalence. Table 6 shows that the Hadamard rotation significantly reduces quantization error, especially under coarser quantization granularity. However, 8-bit quantization of the FC2 input consistently outperforms the rotation, yielding an additional loss reduction of 0.005 to 0.007. This result indicates that 8-bit quantization sets an upper bound on the achievable improvement. Therefore, in our main experiments, we simply set the FC2 input to 8-bit to provide a robust and general baseline for assessing the potential of the W4A4 QAT scaling law, without confounds from activation quantization bottleneck.

