# OpenReview forum: "Scaling Law for Quantization-Aware Training"
_ICLR.cc/2026/Conference — Submitted to ICLR 2026_

### Official Review · Reviewer_csyk · 2025-10-27

**Soundness:** 3
**Presentation:** 2
**Contribution:** 3
**Rating:** 6
**Confidence:** 4

**Summary:**

This work investigate scaling laws a 4-bit precision (W4A4), arguing that exisiting QAT scaling laws often ignore the key factors, such as number of training tokens and quantization granularity. The paper proposes a unified scaling law for QAT that modelling quantiation error as a funciton of model size, size of training data and quantization group size, validated across 268 experiment, uncovering qunatization properties, such as different sensitiviy on weight and activation quantization errors (W16A4, W4A16). Applying mixed-precision quantization to adress bottle necks demonstraing weight and quantization errors can converge to similar levels, descriping how the quantization error can depend on the model size, number of training tokens and quantization granularity (elements in each quant. group). Through experiments this work demonstrates valuable insights for future research in QAT, such as the limitations of activation quantization.

**Strengths:**

1. Demostrating that the quantization error is both dependent on the number of parameters and the training data size for W4A4 QAT.

2. Also including quantization granularity, following prior works.

3. Reproducibility: Models, hyperparameters, dataset (open-source) and evaluation metrics reported.

4. Investigation of weight-only, activation only and both activation and weight quantizations.

5. Demonstrate INT4 is similar or superior in performance to FP4, argues why theoretically and uses it as a rationale for using INT4 quantization in experiments.

6. Learning rate abblation study for W4A4 models, contraring what we have seen reported in BitNet models, arguing the for both uncompressed and QAT training.

7. Generally provides the community with a lot of interesting insights for future QAT research.

**Weaknesses:**

1. Figures 2, 3, 5,6 could benefit for better descriptions making them easier to interpret.

2. Only the Llama3 model family is investigated, potentially openeing a limitation of the scaling law on newer archiectures, limiting the impact.

3. Scaling parameters are fit on 80 runs – Quite a a lot especially for larger models, where compression becomes even more relevant, both for efficiency and performance reasons, limiting the feasible of who can afford to fit the parameters on big models.

**Questions:**

1. Do you expect this to scale beyond 595M parameters?

2. A large number of training tokens for small models, was that necessary to prove the points?

3. The parameters “The parameter \gamma_{N} measures the sensitivity to the model” - Isn’t it more accurate to say it dictates the sensitiveness, as you need quite a lot of runs to fit it to the scale-equations (5). (same for \gamma_{G})

4. Figure 7B demonstrates that W4A16 quantizatione error increases with the number of tokens? I guess it’s the increase in variance in the training data that makes the capacity of the model struggle, but do you have an idea on how sensitive this is? It is within-distribution and what we see is a generalization error, or is the weightage of data being more domain-specific.

---

> ### Author Response · Authors · 2025-11-21
> **Response to Reviewer csyK [1/2]**
>
> Thanks for your thoughtful review that will help us strengthen the manuscript. Below, we address each identified weakness and questions.
>
> **Question 1** : Figures 2, 3, 5,6 could benefit for better descriptions making them easier to interpret.
>
> **Answer 1&#x20;**:  We have revised the captions for Figures 2, 3, 5, and 6 to provide clearer, more informative descriptions in the updated manuscript.
>
>
>
> **Question 2** :  Only the Llama3 model family is investigated, potentially opening a limitation of the scaling law on newer architectures, limiting the impact.
>
> **Answer 2&#x20;**:  We acknowledge this limitation. Our study focuses on Llama3-style dense transformers, which represent the base of many modern LLMs (e.g., Llama, Qwen, Mistral, DeepSeek). To address architecture concerns, we ran preliminary QAT on an MoE model (8 experts out of 40). Due to time and compute limits, we verified trends with respect to N, D, and G. As shown in Table R1, MoE quantization error varies with training tokens and granularity in a way consistent with dense models. We also find that the FC2 input remains the main quantization bottleneck in MoE. These results suggest our scaling law extends to MoE architectures.
>
> **Tables R1**: Experiments on MoE architecture. "74M/301M" indicates 301M total parameters with 74M activate parameters of MoE.
>
> | ID | Model size | Training/Tokens | Precision                 | Quantization error |
> | -- | ---------- | --------------- | ------------------------- | ------------------ |
> | 1  | 74M/301M   | 10B             | W4A4G128                  | 0.067              |
> | 2  | 74M/301M   | 20B             | W4A4G128                  | 0.071              |
> | 3  | 74M/301M   | 50B             | W4A4G128                  | 0.076              |
> | 4  | 74M/301M   | 10B             | W4A4G128                  | 0.067              |
> | 5  | 145M/598M  | 10B             | W4A4G128                  | 0.055              |
> | 6  | 284M/1203M | 10B             | W4A4G128                  | 0.043              |
> | 7  | 74M/301M   | 10B             | W4A4G64                   | 0.052              |
> | 8  | 74M/301M   | 10B             | W4A4G128                  | 0.067              |
> | 9  | 74M/301M   | 10B             | W4A4G256                  | 0.079              |
> | 10 | 74M/301M   | 10B             | W4A4G128                  | 0.067              |
> | 11 | 74M/301M   | 10B             | W4A4G128 (FC2 input 8bit) | 0.044              |
>
>
>
> **Question 3** :  Scaling parameters are fit on 80 runs – Quite a a lot especially for larger models, where compression becomes even more relevant, both for efficiency and performance reasons, limiting the feasible of who can afford to fit the parameters on big models.
>
> **Answer 3&#x20;**:  Our main contribution is the trends of quantization error with N, D, and G, and the separate analysis of activation and weight errors. These insights can guide future quantization algorithms without repeating large sweeps. Regarding the 80 runs, this is a trade-off between cost and reliable extrapolation. More runs increase compute but improve fit stability and predictive accuracy. In practice, fewer runs can work if one accepts larger uncertainty.
>
>
>
> **Question 4** :  Do you expect this to scale beyond 595M parameters?
>
> **Answer 4&#x20;**:  Validating larger models is important. As shown in Figure 5, we trained a \~973M (\~1B) parameter model on 100B and 200B tokens. The scaling law, fit on models up to \~600M parameters, accurately predicts the quantization error for this \~1B model. This supports reliable extrapolation to larger model and data scales.
>
> **Question 5** :  A large number of training tokens for small models, was that necessary to prove the points?
>
> **Answer 5&#x20;**: Yes. Using many training tokens for small models improves the reliability of extrapolation in high data-to-parameter regimes. For example, a 74M model trained on 100B tokens has a data-to-parameter ratio of 1,351. In practice, Llama-3-8B trained on \~16T tokens reaches a ratio near 2,000. Therefore, high token counts for small models are necessary to test and validate the scaling behavior we study.
>
>
>
> **Question 6** : The parameters “The parameter \gamma\_{N} measures the sensitivity to the model” - Isn’t it more accurate to say it dictates the sensitiveness, as you need quite a lot of runs to fit it to the scale-equations (5). (same for \gamma\_{G})
>
> **Answer 6&#x20;**:  Thank you for the suggestion. We have updated the text.
>
>
>
> **Question 7** :  Figure 7B demonstrates that W4A16 quantizatione error increases with the number of tokens?&#x20;
>
> **Answer 7&#x20;**:  Yes. Figure 7B shows that W4A16 quantization error grows with the number of training tokens. This is a key result: the increase is mainly driven by weight quantization. As noted in Question 6, the fitted $\gamma_D$ in Table 1 captures this sensitivity.

---

> ### Author Response · Authors · 2025-11-21
> **Response to Reviewer csyK [2/2]**
>
> **Question 8** : I guess it’s the increase in variance in the training data that makes the capacity of the model struggle, but do you have an idea on how sensitive this is?&#x20;
>
> **Answer 8** :  Prior work (e.g., “Physics of Language Models” \[1] ) shows models can store about 2 bits of knowledge per parameter under BF16/INT8. Moving to 4-bit weights reduces representational capacity further. When D grows, the model must encode more knowledge. Therefore, with low-bit weights, this added knowledge exceeds the available capacity more often, increasing loss and observed quantization error. This aligns with our findings: weight quantization error rises faster with D than activation error because weights must store accumulated knowledge.&#x20;
>
>
>
> \[1] Physics of Language Models: Part 3.3, Knowledge Capacity Scaling Laws, https://arxiv.org/abs/2404.05405
>
>
>
> **Question 9** : It is within-distribution and what we see is a generalization error, or is the weightage of data being more domain-specific.
>
> **Answer 9** : The pretraining data is randomly shuffled and spans multiple domains. Therefore, the observed quantization error reflects generalization within the mixed distribution, not domain-specific effects.
>
> We sincerely appreciate the time and efforts you have dedicated to reviewing our paper. Should you have any further inquiries, please let us know and we are more than delighted to discuss with you.

---

### Official Review · Reviewer_Xd87 · 2025-10-28

**Soundness:** 3
**Presentation:** 3
**Contribution:** 2
**Rating:** 4
**Confidence:** 3

**Summary:**

This paper investigates how quantization-aware training (QAT) scales with model size, dataset size, and quantization granularity. Through 268 from-scratch QAT experiments on LLaMA-style models, they empirically show that quantization error decreases with larger models, but increases with larger datasets and coarser quantization groups. They further decompose the error into weight and activation components and identify the FC2 layer as a key activation outlier bottleneck, suggesting that mixed-precision treatment (8-bit for FC2) can balance both sources of error.

**Strengths:**

- Presents the first unified QAT scaling law incorporating model size (N), dataset size (D), and quantization granularity (G).
- Provides clear empirical validation with extensive experiments and well-fitted results
- Offers useful diagnostic insights, including weight vs. activation error decomposition and identification of the FC2 activation bottleneck.

**Weaknesses:**

- The definition of quantization error differs from conventional quantization studies. In this paper, the “quantization error” is defined as the final training loss gap rather than the difference between quantized and full-precision parameters or inference performance degradation.
There is no clear rationale for this definition, and the paper does not show whether a smaller loss gap actually correlates with better downstream performance when compared to BF16-trained counterparts.
This makes it difficult to interpret the analysis in a practically meaningful way.
- The paradigm of QAT used in this paper also deviates from typical practice. In most real-world scenarios, QAT is applied on well-trained full-precision LLMs to recover accuracy after quantization, whereas this paper applies QAT from scratch. It is unclear whether a model trained from scratch under 4-bit quantization can achieve comparable or better performance than a full-precision model followed by QAT.
Moreover, the models studied are mostly below 1B parameters, which further limits the generality of the conclusions.
- The depth of contribution appears somewhat limited. Extending the scaling law to include additional factors (dataset size and quantization granularity) is a valuable analytical contribution, but the idea of mitigating outlier activation errors by applying higher precision to specific layers has already been extensively discussed in prior mixed-precision quantization works.

**Questions:**

See weakness

---

> ### Author Response · Authors · 2025-11-21
> **Response to Reviewer Xd87 [1/2]**
>
> Thanks for your thoughtful review that will help us strengthen the manuscript. Below, we address each identified weakness and questions.
>
> **Question 1** :  The definition of quantization error differs from conventional quantization studies. In this paper, the “quantization error” is defined as the final training loss gap rather than the difference between quantized and full-precision parameters or inference performance degradation. There is no clear rationale for this definition, and the paper does not show whether a smaller loss gap actually correlates with better downstream performance when compared to BF16-trained counterparts. This makes it difficult to interpret the analysis in a practically meaningful way.
>
> **Answer 1&#x20;**:  In the scaling-law literature, it is standard to model final training/validation loss rather than downstream metrics. Lower loss is widely used as a proxy for better downstream performance \[1–5]. Following this practice, we define “quantization error” as the final loss gap. Moreover, Table R1 shows a strong correlation between loss gap and downstream accuracy in our experiments, supporting the practical relevance of this definition.
>
>
>
> Table R1: Loss gap and downstream task accuracy of 973M models trained with 100B tokens across different quantization groups. "Avg Accuracy" indicates the 5-shot average accuracy  across WinoGrande, HellaSwag, Arc\_Challenge, Arc\_Easy and PIQA.
>
> | Precision | Groups  | Quantization Error (loss Gap) | Avg Accuracy |
> | --------- | ------- | ----------------------------- | ------------ |
> | BF16      | -       | 0                             | 60.69        |
> | W4A4      | -1      | 0.077                         | 57.62        |
> | W4A4      | 256     | 0.067                         | 58.41        |
> | W4A4      | 128     | 0.060                         | 58.83        |
> | W4A4      | 64      | 0.048                         | 59.23        |
> | W4A4      | 32      | 0.040                         | 59.58        |
>
>
>
> \[1] Precision Scaling Law, ICLR 2025
>
> \[2] Training compute-optimal large language models, NeurIPS 2022
>
> \[3] Compression Scaling Laws: Unifying Sparsity and Quantization
>
> \[4] Scaling laws for neural language models.
>
> \[5] Unified Scaling Laws for Compressed Representations
>
>
>
> **Question 2** :  The paradigm of QAT used in this paper also deviates from typical practice. In most real-world scenarios, QAT is applied to well-trained full-precision LLMs to recover accuracy after quantization, whereas this paper applies QAT from scratch. It is unclear whether a model trained from scratch under 4-bit quantization can achieve comparable or better performance than a full-precision model followed by QAT. Moreover, the models studied are mostly below 1B parameters, which further limits the generality of the conclusions.
>
> **Answer 2&#x20;**:  We agree that QAT based on existing pretrained models is a more **cost-efficient** approach and more practical. However, our work focuses on a different but equally important goal: achieving the **best possible performance** with low-bit models. There are two key reasons why from-scratch 4-bit QAT is a practical and relevant research area:
>
> 1\. The industry is moving towards 4-bit computation. For instance, NVIDIA's new Blackwell GPU architecture includes native support for FP4 precision. As 8-bit training now achieves nearly lossless performance compared to full precision, 4-bit training is the next frontier for accelerating large-scale model development. Our work on from-scratch W4A4 Quantization-Aware Training (QAT) provides a foundational understanding of the scaling laws that will govern this future.  Therefore, from-scratch 4-bit training is foundational for **accelerating the training of future LLMs** from the ground up, not just for inference.
>
> 2\. When maximum performance is the priority, from-scratch training still holds an advantage. Our new experiments in Table R2 compare QAT from a pretrained checkpoint versus QAT from scratch, given the same training token budget. The results show that while fine-tuning an existing pretrained model is efficient, **training from scratch achieves a lower final loss**. Therefore, for real-world scenarios where achieving state-of-the-art performance is critical, from-scratch QAT remains a vital and necessary technique.
>
>
>
> **Table R2**: W4A4G64 145M models with 20B training tokens.
>
> | Model                  | Final loss |
> | ---------------------- | ---------- |
> | 100% BF16 (Baseline)   | 3.125      |
> | 80% BF16 + 20% W4A4G64 | 3.211      |
> | 60% BF16 + 40% W4A4G64 | 3.201      |
> | 40% BF16 + 60% W4A4G64 | 3.196      |
> | 20% BF16 + 80% W4A4G64 | 3.193      |
> | 0% BF16 + 100% W4A4G64 | **3.190**  |

---

> ### Author Response · Authors · 2025-11-21
> **Response to Reviewer Xd87 [2/2]**
>
> **Question 3** :  The depth of contribution appears somewhat limited. Extending the scaling law to include additional factors (dataset size and quantization granularity) is a valuable analytical contribution, but the idea of mitigating outlier activation errors by applying higher precision to specific layers has already been extensively discussed in prior mixed-precision quantization works.
>
> **Answer 3&#x20;**: Mixed precision is not our core contribution. Our key finding is that, even with QAT, the FC2 layers remain the main bottleneck for activation quantization error. These layers should use higher precision or outlier-mitigation methods. We use mixed precision mainly to study the new error distribution after addressing the FC2 activation bottleneck. With the FC2 activation issue reduced, we find that weight quantization error can become comparable to, or even larger than, activation error. This highlights the need to study weight quantization in future more carefully, which has been largely overlooked. **In summary, our contribution is identifying the FC2 activation bottleneck and showing the rising importance of weight quantization error once that bottleneck is addressed.**
>
>
> We sincerely appreciate the time and efforts you have dedicated to reviewing our paper. Should you have any further inquiries, please let us know and we are more than delighted to discuss with you.

---

> ### Author Response · Authors · 2025-11-27
> **Looking forward to the follow-up discussion**
>
> Dear Reviewer Xd87,
>
> We would like to express our sincere appreciation for the time and effort you have dedicated to reviewing our paper.
>
> We hope our rebuttal has adequately addressed your concerns. Please let us know if you have any further questions. We would be happy to provide additional clarifications.
>
> Best regards,
>
> The Authors

---

### Official Review · Reviewer_ff79 · 2025-11-03

**Soundness:** 3
**Presentation:** 3
**Contribution:** 3
**Rating:** 6
**Confidence:** 3

**Summary:**

This paper proposes a unified scaling law for quantization-aware training (QAT) of large language models, linking quantization error to model size, training tokens, and quantization granularity. Across 268 experiments on LLaMA-style models, the authors find that error decreases with model size but grows with more data and coarser quantization. They identify activation outliers in the FC2 layer as the main 4-bit bottleneck and show mixed-precision (8-bit FC2) largely fixes it. The study provides clear empirical and theoretical insights for improving low-bit LLM training.

**Strengths:**

1. The unified scaling law feels intuitive yet backed by solid data; the inclusion of both data and granularity terms makes sense.
2. Clear identification of the FC2 activation bottleneck; the mixed-precision fix is simple and convincing.
3. This paper is a neat empirical work that connects scaling laws and quantization in a meaningful, practically useful way.

**Weaknesses:**

1. Experiments stop at sub-1B dense models — unclear if the scaling law still holds for >10B or MoE setups.
2. Mostly focuses on W4A4; doesn’t explore ternary or mixed-bit cases that recent works care about.
3. While empirical results are strong, the practical takeaways for real deployment (beyond FC2 mixed precision) could be discussed more.

**Questions:**

1. The paper argues that quantization error increases with more training tokens — which is counterintuitive from a generalization standpoint. Could the authors provide a deeper explanation or theoretical intuition for why additional data worsens quantization sensitivity?
2. The scaling exponents ($\gamma_N$, $\gamma_D$ and $\gamma_G$) are empirically fitted. Do they exhibit any consistency across model families or datasets, suggesting a more universal law, or are they dataset-dependent heuristics?

---

> ### Author Response · Authors · 2025-11-21
> **Response to Reviewer ff79 [1/2]**
>
> Thanks for your thoughtful review that will help us strengthen the manuscript. Below, we address each identified weakness and questions.
>
> **Question 1** :  Experiments stop at sub-1B dense models — unclear if the scaling law still holds for >10B or MoE setups.
>
> **Answer 1&#x20;**:   Our quantization error trends appear general across scales. The largest model we tested (973M) is 13× larger than the smallest (74M), and the trend holds across this range, suggesting it may extend to >10B. We could not run QAT on >10B models due to compute limits. **We also ran preliminary QAT on MoE models (8 experts of 40).** As summarized in Table R1, quantization error varies with N, D, and G in the same way as in dense models, and the FC2 input remains the main activation bottleneck. These results indicate our scaling law likely extends to MoE architectures.
>
> **Tables R1**: Experiments on MoE architecture. "74M/301M" indicates 301M total parameters with 74M activate parameters of MoE.
>
> | ID | Model size | Training/Tokens | Precision                 | Quantization error |
> | -- | ---------- | --------------- | ------------------------- | ------------------ |
> | 1  | 74M/301M   | 10B             | W4A4G128                  | 0.067              |
> | 2  | 74M/301M   | 20B             | W4A4G128                  | 0.071              |
> | 3  | 74M/301M   | 50B             | W4A4G128                  | 0.076              |
> | 4  | 74M/301M   | 10B             | W4A4G128                  | 0.067              |
> | 5  | 145M/598M  | 10B             | W4A4G128                  | 0.055              |
> | 6  | 284M/1203M | 10B             | W4A4G128                  | 0.043              |
> | 7  | 74M/301M   | 10B             | W4A4G64                   | 0.052              |
> | 8  | 74M/301M   | 10B             | W4A4G128                  | 0.067              |
> | 9  | 74M/301M   | 10B             | W4A4G256                  | 0.079              |
> | 10 | 74M/301M   | 10B             | W4A4G128                  | 0.067              |
> | 11 | 74M/301M   | 10B             | W4A4G128 (FC2 input 8bit) | 0.044              |
>
>
>
> **Question 2** :  Mostly focuses on W4A4; doesn’t explore ternary or mixed-bit cases that recent works care about.
>
> **Answer 2&#x20;**: Thank you for your valuable feedback. We agree that exploring various bit-widths is an important research direction. Our study deliberately focuses on **4-bit quantization** for the following reasons:
>
> * **8-bit quantization** generally achieves near-lossless performance, making it less critical to analyze for scaling law errors.
>
> * **5, 6, and 7-bit formats** lack native hardware support, which limits their practical application.
>
> * **4-bit quantization** represents a practical and widely used bit-width where quantization error becomes a significant factor.
>
> Given that **4-bit quantization already introduces substantial error,** we chose to first conduct an in-depth analysis of its bottlenecks (i.e., separating weight and activation errors) rather than moving to more extreme bit-widths like 2-bit or ternary quantization. This focused approach allows us to provide a clear baseline for understanding quantization effects **in practical scenarios.**
>
>
>
> **Question 3** :  While empirical results are strong, the practical takeaways for real deployment (beyond FC2 mixed precision) could be discussed more.
>
> **Answer 3**: Thank you for the suggestion. The practical takeaways from our work are:
>
> * &#x20;The FC2 layer input should use higher precision or outlier-mitigation methods, because QAT keeps strong outliers in this layer.
>
> * &#x20;In practice, teams often test quantization on small models or small datasets. Our scaling law shows that quantization error depends on the data-to-parameters ratio. We recommend testing across multiple ratios to get robust conclusions, rather than relying on a single scale.
>
> * &#x20;The community mainly focuses on activation quantization error. After removing the FC2 Proj input bottleneck, further improvements to W4A4 QAT should optimize both weight and activation quantization together. Their effects become similar, and weight quantization can even exceed activation quantization when the data-to-parameters ratio is above 500 (see Figure 8). This suggests shifting from activation-only methods to approaches that balance both errors.
>
> * &#x20;Quantization group size has a strong impact on error. For deployment, use more fine-grained formats, such as MXFP8 and NVFP4, instead of traditional FP8 and FP4.

---

> ### Author Response · Authors · 2025-11-21
> **Response to Reviewer ff79 [2/2]**
>
> **Question 4** :  The paper argues that quantization error increases with more training tokens — which is counterintuitive from a generalization standpoint. Could the authors provide a deeper explanation or theoretical intuition for why additional data worsens quantization sensitivity?
>
> **Answer 4&#x20;**:   Our results show that W4A4 quantization error grows as the number of training tokens increases, and Section 4.3 indicates the main source is weight quantization (W4A16). Intuitively, pretraining stores information from data into the model’s weights.  Prior work (e.g., “Physics of Language Models” \[1] ) shows models can store about 2 bits of knowledge per parameter under BF16/INT8. Moving to 4-bit weights reduces representational capacity further. When D grows, the model must encode more knowledge. Therefore, with low-bit weights, this added knowledge exceeds the available capacity more often, increasing loss and observed quantization error. This aligns with our findings: weight quantization error rises faster with D than activation error because weights must store accumulated knowledge.&#x20;
>
>
>
> \[1] Physics of Language Models: Part 3.3, Knowledge Capacity Scaling Laws, https://arxiv.org/abs/2404.05405
>
>
>
> **Question 5** :  The scaling exponents ($\gamma$) are empirically fitted. Do they exhibit any consistency across model families or datasets, suggesting a more universal law, or are they dataset-dependent heuristics?
>
> **Answer 5&#x20;**:   As prior scaling studies \[1,2] show, the fitted exponent ($\gamma$) depends strongly on both model architecture and dataset, and thus varies across setups. Therefore, we do not view $\gamma$ as universal. Instead, the consistent trends we observe and our error analysis are the main takeaways.
>
>
>
> \[1] Precision Scaling Law, ICLR 2025
>
> \[2]  Training compute-optimal large language models, NeurIPS 2022
>
> We sincerely appreciate the time and efforts you have dedicated to reviewing our paper. Should you have any further inquiries, please let us know and we are more than delighted to discuss with you.

---

### Official Review · Reviewer_G79L · 2025-11-08

**Soundness:** 3
**Presentation:** 2
**Contribution:** 1
**Rating:** 4
**Confidence:** 4

**Summary:**

This paper proposes a unified scaling law for 4-bit Quantization-Aware Training (W4A4) in language models. The authors model the quantization error as a function of not only model size but also the number of training tokens and the quantization group size. Based on 268 experiments, they formulate $\delta_p \propto D^{\gamma_D} G^{\gamma_G} / N^{\gamma_N}$, finding that error decreases with larger models but, counter-intuitively, increases with more training data. The paper further decomposes this error, identifies activation quantization as the primary bottleneck, and pinpoints it to outliers in the FC2 projection layer's input.

**Strengths:**

- Significance: The work addresses a critical and timely problem. As W4A4 QAT becomes essential for efficient LLM deployment, understanding its scaling behavior is of high practical importance, yet it remains poorly understood.

- Originality: The primary originality lies in the formulation of a scaling law that includes the training data volume (D) and quantization granularity (G). The finding that quantization error increases with D is a novel and non-trivial observation that challenges common assumptions.

- Quality: The study is methodologically sound, built upon a substantial empirical foundation of 268 QAT experiments. The approach of decomposing the total error into weight-only ($\delta_{W4A16}$) and activation-only ($\delta_{W16A4}$) components (Fig. 6) is a systematic way to investigate the error sources.

- Clarity: The paper is well-written, and the core experimental results (Fig. 4, Fig. 7) are presented clearly, making the authors' main observations easy to follow.

**Weaknesses:**

- Limited Generalizability Due to Model Scale: The paper's most significant weakness is the gap between its claims ("Large Language Models") and its experimental setup. The scaling law is derived from models ranging from 74M to 595M parameters, with validation on a 973M model. These models are orders of magnitude smaller than current state-of-the-art LLMs (e.g., 7B, 70B, 100B+). Scaling laws are only valuable if they extrapolate, and there is no evidence that a trend observed in sub-1B models will hold for models 100x or 1000x larger, which may exhibit different emergent properties and quantization sensitivities. This discrepancy severely limits the trustworthiness and practical applicability of the proposed law for actual LLMs.

- Lack of Mechanistic Analysis for the Effect of D: A key finding of the paper is that quantization error $\delta_p$ increases with the data volume D. The authors observe this (Fig. 4b) and note that weight quantization error ($\delta_{W4A16}$) is more sensitive to D than activation error (Table 1, $\gamma_D$ values). However, the paper stops at this observation and fails to provide a deeper, mechanistic explanation for why this occurs. Does training on more data fundamentally increase the complexity of the learned weight representations, making them harder to compress into 4 bits? Does it alter the weight distributions or introduce more outliers? This lack of causal analysis turns a potentially profound discovery into a superficial empirical note.

- Insufficient Novelty: The identification of the FC2 projection input (following the SwiGLU non-linearity) as the primary source of activation outliers (Fig. 9a) is not a new discovery. This has been a well-documented phenomenon in the Post-Training Quantization (PTQ) literature for years (e.g., in work like SmoothQuant). The paper confirms this finding holds for QAT but does not offer a new insight specific to the QAT process itself.

- Trivial Solution for Bottleneck Analysis: The proposed "solution" to this bottleneck is to use 8-bit precision for the FC2 input (Fig. 9b). This mixed-precision approach is a workaround that avoids the core W4A4 challenge rather than solving it. This analysis does not inform how to improve a true W4A4 model. A more valuable contribution would have involved exploring QAT-native techniques (e.g., targeted regularization, learnable clipping functions for outliers) that operate within the W4A4 constraint.

**Questions:**

Fundamentally, why do the authors think the loss gets worse as the training data (D) increases?

It is counterintuitive, since the primary mechanism of QAT is to robustify the model against quantization error throughout training; thus, longer training is expected to close the accuracy gap between the full-precision and the quantized model. This paper's observation could be contradicted by the practical successes of reduced-precision inference (e.g., NVIDIA NVFP4 - https://developer.nvidia.com/blog/introducing-nvfp4-for-efficient-and-accurate-low-precision-inference/, Apple Quality-Recovery Adapter - https://arxiv.org/pdf/2507.13575v1) that incorporate billion-token finetuning to fully recover the FP model's accuracy.

Can the authors clarify why their observation of loss degradation with increased training data (Fig. 4(b)) makes sense?  One possible sync-up point is as follows: Scaling Laws for Precision (https://arxiv.org/pdf/2411.04330) and ParetoQ (https://arxiv.org/pdf/2502.02631) show that an intermediate training budget helps achieve pareto-optimal QAT accuracy when aggressive quantization is applied. It would be helpful to scale the training tokens in Fig. 4(b) to a lower budget (1M-1B tokens) to see if a consistent trend emerges.

[Willing to update the ratings based on this discussion]

---

> ### Author Response · Authors · 2025-11-21
> **Response to Reviewer G79L [1/2]**
>
> Thanks for your thoughtful review that will help us strengthen the manuscript. Below, we address each identified weakness and questions.
>
> **Question1** : Limited Generalizability Due to Model Scale: The paper's most significant weakness is the gap between its claims ("Large Language Models") and its experimental setup. The scaling law is derived from models ranging from 74M to 595M parameters, with validation on a 973M model. These models are orders of magnitude smaller than current state-of-the-art LLMs (e.g., 7B, 70B, 100B+). Scaling laws are only valuable if they extrapolate, and there is no evidence that a trend observed in sub-1B models will hold for models 100x or 1000x larger, which may exhibit different emergent properties and quantization sensitivities. This discrepancy severely limits the trustworthiness and practical applicability of the proposed law for actual LLMs.
>
> **Answer 1** :   Our quantization error trends and analysis appear general across scales. Our largest tested model (973M) is 13× larger than the smallest (74M), and the trend remains consistent across this range. This suggests the behavior is likely to hold with further 10× scale-ups (>10B). That said, we could not run full QAT on >10B models due to prohibitive compute costs, so we acknowledge this as a limitation.
>
>
>
> **Question2** :  Lack of Mechanistic Analysis for the Effect of D: A key finding of the paper is that quantization error  $\delta_p$ increases with the data volume D. The authors observe this (Fig. 4b) and note that weight quantization error ($\delta_{W4A16}$) is more sensitive to D than activation error (Table 1,  $\gamma_D$values). However, the paper stops at this observation and fails to provide a deeper, mechanistic explanation for why this occurs. Does training on more data fundamentally increase the complexity of the learned weight representations, making them harder to compress into 4 bits? Does it alter the weight distributions or introduce more outliers? This lack of causal analysis turns a potentially profound discovery into a superficial empirical note.
>
> **Answer2** :  Prior work (e.g., “Physics of Language Models” \[1] ) shows models can store about 2 bits of knowledge per parameter under BF16/INT8. Moving to 4-bit weights reduces representational capacity further. When D grows, the model must encode more knowledge. Therefore, with low-bit weights, this added knowledge exceeds the available capacity more often, increasing loss and observed quantization error. This aligns with our findings: weight quantization error rises faster with D than activation error because weights must store accumulated knowledge.&#x20;
>
>
>
> \[1] Physics of Language Models: Part 3.3, Knowledge Capacity Scaling Laws, https://arxiv.org/abs/2404.05405
>
>
>
> **Question3** :  Insufficient Novelty: The identification of the FC2 projection input (following the SwiGLU non-linearity) as the primary source of activation outliers (Fig. 9a) is not a new discovery. This has been a well-documented phenomenon in the Post-Training Quantization (PTQ) literature for years (e.g., in work like SmoothQuant). The paper confirms this finding holds for QAT but does not offer a new insight specific to the QAT process itself.
>
> **Answer3** : Our contribution is specific to QAT. In PTQ, activation outliers are widespread across many layers (as shown by works like SmoothQuant), and mitigation is applied broadly. In contrast, our results show that QAT itself suppresses outliers in several key layers: QKV Proj, O Proj, and FC1 Proj. As stated in Sec. 4.4 (lines 429–431), these layers no longer need extra outlier handling after QAT. Only the FC2 layer remains a dominant source of activation outliers under QAT, which is why targeted mitigation there is effective. This narrows the problem and provides a QAT-specific insight beyond prior PTQ observations.
>
>
>
> **Question4** :  Trivial Solution for Bottleneck Analysis: The proposed "solution" to this bottleneck is to use 8-bit precision for the FC2 input (Fig. 9b). This mixed-precision approach is a workaround that avoids the core W4A4 challenge rather than solving it. This analysis does not inform how to improve a true W4A4 model. A more valuable contribution would have involved exploring QAT-native techniques (e.g., targeted regularization, learnable clipping functions for outliers) that operate within the W4A4 constraint.
>
> **Answer4** : We tested both 8-bit precision and random Hadamard rotation (see Table 6 in Appendix). We chose 8-bit for its simplicity and stronger results. As noted (lines 455–457 in paper), 8-bit serves as an upper bound, giving a clear baseline to isolate the FC2 input bottleneck. Once this bottleneck is removed, W4A4 QAT improvements depend on jointly reducing weight and activation errors, as their impacts become similar. This insight guides future work toward QAT-native methods within W4A4 , focusing on balanced optimization rather than only currently activation handling.

---

> ### Author Response · Authors · 2025-11-21
> **Response to Reviewer G79L [2/2]**
>
> **Question5** :  Fundamentally, why do the authors think the loss gets worse as the training data (D) increases? It is counterintuitive, since the primary mechanism of QAT is to robustify the model against quantization error throughout training; thus, longer training is expected to close the accuracy gap between the full-precision and the quantized model. This paper's observation could be contradicted by the practical successes of reduced-precision inference (e.g., NVIDIA NVFP4 - <https://developer.nvidia.com/blog/introducing-nvfp4-for-efficient-and-accurate-low-precision-inference/>, Apple Quality-Recovery Adapter - <https://arxiv.org/pdf/2507.13575v1>) that incorporate billion-token finetuning to fully recover the FP model's accuracy. Can the authors clarify why their observation of loss degradation with increased training data (Fig. 4(b)) makes sense?
>
> **Answer5** :  As noted in Answer 2, each parameter has limited capacity to store knowledge; lower-bit weights reduce this capacity further. When D increases, the model must encode more knowledge. Under W4A4, this added knowledge exceeds the representational budget more often, widening the loss gap versus BF16, even with QAT.
>
> Additionally, **this does not conflict with reports of strong low-precision inference** after long finetuning (e.g., NVFP4 \[2]).  In fact, NVFP4 training results (Fig. 6a in paper \[2], official paper of NVFP4 training) show the loss gap versus BF16 grows with more data (e.g., from \~0.5% at 0.8T to \~1.5% at 1T during cosing learning rate decay), consistent with our observation. Our finding highlights a real capacity-pressure effect under strict low-bit constraints: more data can improve both models, but the quantized model improves less when its representational limit is the bottleneck.&#x20;
>
>
>
> \[2] Pretraining Large Language Models with NVFP4,https://arxiv.org/abs/2509.25149v1
>
>
>
> **Question6** :  One possible sync-up point is as follows: Scaling Laws for Precision (https://arxiv.org/pdf/2411.04330) and ParetoQ (https://arxiv.org/pdf/2502.02631) show that an intermediate training budget helps achieve pareto-optimal QAT accuracy when aggressive quantization is applied.&#x20;
>
> **Answer6** :  We focus on maximizing performance for strict low-bit models. This complements works showing Pareto trade-offs under limited budgets.
>
> * Industry trajectory: Hardware is moving to 4-bit (e.g., NVIDIA Blackwell FP4). As 8-bit training is near-lossless, 4-bit is the next step. From-scratch W4A4 QAT helps establish the scaling rules needed for this shift.
>
> * Performance priority: With the same token budget, our new results (Table R1) show from-scratch QAT reaches lower final loss than QAT from a pretrained checkpoint. Fine-tuning is efficient, but training from scratch achieves better end performance.
>
> Thus, our aim is to understand and push the best achievable accuracy under strict W4A4, which is key for future 4-bit training regimes.
>
>
>
> **Table R1**: W4A4G64 145M models with 20B training tokens.
>
> | Model                  | Final loss |
> | ---------------------- | ---------- |
> | 100% BF16 (Baseline)   | 3.125      |
> | 80% BF16 + 20% W4A4G64 | 3.211      |
> | 60% BF16 + 40% W4A4G64 | 3.201      |
> | 40% BF16 + 60% W4A4G64 | 3.196      |
> | 20% BF16 + 80% W4A4G64 | 3.193      |
> | 0% BF16 + 100% W4A4G64 | **3.190**  |
>
>
>
> **Question 7** : It would be helpful to scale the training tokens in Fig. 4(b) to a lower budget (1M-1B tokens) to see if a consistent trend emerges.
>
> **Answer 7** :  We extended Fig. 4(b) with a 1B-token setting (Table R2). The trend is consistent: the loss gap vs. BF16 is smallest at 1B and grows with more tokens. This supports our claim that the gap widens as D increases under strict low-bit constraints.
>
>
>
> Table R2: Figure 4(b) results with additional 1B training token results.
>
> | Group Size | Training Tokens | Loss Gap  |
> | ---------- | --------------- | --------- |
> | 256        | 1B              | **0.044** |
> | 256        | 10B             | 0.054     |
> | 256        | 20B             | 0.056     |
> | 256        | 50B             | 0.061     |
> | 256        | 100B            | 0.063     |
> | 32         | 1B              | **0.027** |
> | 32         | 10B             | 0.03      |
> | 32         | 20B             | 0.031     |
> | 32         | 50B             | 0.037     |
> | 32         | 100B            | 0.039     |
>
>
> We sincerely appreciate the time and efforts you have dedicated to reviewing our paper. Should you have any further inquiries, please let us know and we are more than delighted to discuss with you.

---

> ### Author Response · Authors · 2025-11-27
> **Looking forward to the follow-up discussion**
>
> Dear Reviewer G79L,
>
> We would like to express our sincere appreciation for the time and effort you have dedicated to reviewing our paper.
>
> We hope our rebuttal has adequately addressed your concerns. Please let us know if you have any further questions. We would be happy to provide additional clarifications.
>
> Best regards,
>
> The Authors

---

### Author Response · Authors · 2025-12-01
**Summary of Rebuttal: Concise post-rebuttal highlights and resolution**

We thank all the reviewers and area chair for their valuable feedback and great efforts. Below we summarize reviewer-positive feedback and how our rebuttal addresses the raised weaknesses, with explicit reviewer references.
## Core contributions (endorsed across reviews)
- Unified QAT scaling law modeling quantization error as a function of model size (N), training tokens (D), and quantization granularity (G).
- Error decomposition into weight vs. activation components, showing differential sensitivities and a QAT-specific activation bottleneck at FC2.
- Actionable guidance: mitigate FC2 outliers; evaluate across data-to-parameter ratios; after FC2 mitigation, jointly optimize weight and activation errors; prefer fine-grained groupings.

## Reviewer positives
- **G79L**: Timely, important problem; originality in including D and G; strong empirical base (268 runs); clear core results and decomposition.
- **ff79**: Unified law is intuitive and well supported; FC2 bottleneck identification and fix are convincing; practical connection between scaling laws and quantization.
- **Xd87**: First unified QAT scaling law covering N/D/G; extensive validation; useful diagnostics (weight vs. activation; FC2 bottleneck).
- **csyk**: Solid insights on dependence on N/D/G; reproducibility; thorough ablations (INT4 vs FP4, LR); guidance valuable for future QAT research.

## Main concerns and how we addressed them
- Generalization to >10B and MoE (G79L, ff79, csyk)
  - We show trend stability from 74M to 973M (13×). Fits up to ~600M accurately predict ~1B (Figure 5). Preliminary MoE results (8-of-40 experts) exhibit the same N/D/G dependencies and FC2 bottleneck. We acknowledge compute limits for >10B.

- Mechanism for error increasing with D (G79L, ff79, csyk)
  - Capacity-based explanation: low-bit weights reduce representational capacity; larger D requires encoding more knowledge in weights, increasing the loss gap. This aligns with “Physics of LMs” capacity scaling and NVFP4 training curves showing widening BF16–low-bit gaps with more data.

- Novelty vs PTQ FC2 findings (G79L, Xd87)
  - QAT-specific insight: QAT suppresses activation outliers in several layers (QKV, O, FC1), uniquely leaving FC2 as the dominant bottleneck, which narrows mitigation under QAT.

- Mixed-precision seen as workaround (G79L)
  - We used 8-bit at FC2 as an upper bound to isolate the bottleneck; tested alternatives (Hadamard). Crucially, after FC2 mitigation, weight and activation errors become comparable, directing future work to W4A4-native joint optimization.

- Quantization error defined as final loss gap; downstream relevance (Xd87)
  - Consistent with scaling-law practice; we provide empirical correlation between reduced loss gap and better downstream accuracy.

- From-scratch QAT vs finetune paradigm (Xd87)
  - From-scratch W4A4 is foundational for emerging native 4-bit training (e.g., NVIDIA Blackwell FP4). At equal token budgets, from-scratch QAT achieves lower final loss than switching from BF16 later, supporting maximum-performance regimes.

- Scope and clarity (ff79, csyk)
  - Focus on 4-bit due to hardware relevance and non-trivial errors.

## Summary
- Multiple reviewers find the work sound and useful. The rebuttal provides mechanistic intuition, external corroboration, extrapolation evidence to ~1B and MoE, downstream correlation for the metric, and clear guidance for W4A4-native follow-ups.
- Overall, the paper delivers a substantive, empirically grounded QAT scaling law, inspiring the future development of QAT algorithm.

---

### Meta-Review · Area_Chair_dcUH · 2026-01-03

**Summary:**

On the positive side, reviewers recognized the paper’s substantial empirical effort (268 QAT runs) and the clarity of its proposed formulation linking quantization error to model size, data volume, and quantization granularity. However, the paper's biggest issue is that the results do not reflect true LLM scaling, which undermines the “LLM scaling law” claim. Since the authors explicitly choose LLM scaling law as the topic, it is fair game to ask for true LLMs in the order of tens of billions of parameters. This was not resolved. Secondly, there is a sigifnicant lack of mechanistic foundations that explain the observations. As framed, the conclusions risk being overly specific rather than broadly predictive. This is also not appropriately addressed via the rebuttal.

**Reviewer Concerns:**

I think the rebuttal addressed several **secondary** concerns, including better empirical justification for the dependence of quantization error on training tokens, additional ablations (e.g., weight vs. activation error decomposition), and clarifications on how the proposed formulation differs from prior PTQ-focused analyses. However, the **major** concerns remain outstanding. In particular, the rebuttal does not resolve the lack of experiments on true LLM scales (tens of billions of parameters), which is critical given the paper’s explicit framing around LLM scaling laws. In addition, while the authors provided intuitive explanations, the rebuttal does not establish a sufficiently deep mechanistic foundation to ensure that the observed trends are general rather than specific to sub-1B models, W4A4 QAT, and the chosen training setup.

**Reviewer Scores:**

The major issues were not resolved, so I am not convinced that the scores would be improved.

---

### Decision · Program_Chairs · 2026-01-26

Reject